# Expansion quantization network: A micro-emotion detection and annotation framework

Jingyi Zhou[1¤*], Senlin Luo[1*], Haofan Chen[2]

**1** School of Information and Electronics, Beijing Institute of Technology, Beijing, China, **2** China Electronics Engineering Design Institute Co., Ltd., Beijing, China

¤ Current address: Institute of Scientific and Technical Research on Archives, Beijing, China
* annezjy94@163.com (JYZ); luosenlin2019@126.com (SLL)

## Abstract

Textemotion detection constitutes a crucial foundation for advancing artificial intelligence from basic comprehension to the exploration of emotional reasoning. Most existing emotion detection datasets rely on manual annotations, which are associated with high costs, substantial subjectivity, and severe label imbalances. This is particularly evident in the inadequate annotation of micro-emotions and the absence of emotional intensity representation, which fail to capture the rich emotions embedded in sentences and adversely affect the quality of downstream task completion. By proposing an all-labels and training-set label regression method, we map label values to energy intensity levels, thereby fully leveraging the learning capabilities of machine models and the interdependencies among labels to uncover multiple emotions within samples. This led to the establishment of the Emotion Quantization Network (EQN) framework for micro-emotion detection and annotation. Using five commonly employed sentiment datasets, we conducted comparative experiments with various models, validating the broad applicability of our framework within NLP machine learning models. Based on the EQN framework, emotion detection and annotation are conducted on the GoEmotions dataset. A comprehensive comparison with the results from its literature demonstrates that the EQN framework possesses a high capability for automatic detection and annotation of micro-emotions. The EQN framework is the first to achieve automatic micro-emotion annotation with energy-level scores, providing strong support for further emotion detection analysis and the quantitative research of emotion computing.

## Introduction

Emotions represent one of the most intricate intrinsic experiences of humanity. Artificial intelligence can gain profound insights into and analyze human emotions through emotion detection, allowing for a better understanding of people's emotional responses

**Data availability statement:** This project is available on Figshare at the following DOI: 10.6084/m9.figshare.30406315.

**Funding:** This research is partially supported by the 242 National Information Security Projects,PR China under Grant 2020A065. The funders had no role in study design, data collection and analysis, decision to publish, or preparation of the manuscript.

**Competing interests:** The authors have declared that no competing interests exist.

[1]. This capability can help individuals comprehend their motivations, enhance the rationality of decision-making, improve interpersonal relationships [2], treat psychological disorders such as depression and anxiety [3], and develop robots with greater emotional understanding to enhance user experiences, among other benefits.

Emotion datasets are a vital resource in NLP research. Currently, the most widely used datasets are single-label datasets [4–7], where each sample can belong to only one emotional category. Traditional single-label emotion datasets have ranged from early binary or ternary categories (such as "positive," "negative," and "neutral") [8] to the six basic emotions of joy, sadness, anger, fear, disgust, and surprise proposed by Ekman [9], and Plutchik's eight basic emotions [10].Single-label datasets are relatively straightforward to annotate, incur lower manual costs, and are less prone to errors. However, a single sentence or passage often contains multiple emotions, which cannot be adequately summarized with simple labels, leading to the emergence of multi-label emotion datasets. In multi-label datasets, each sample can simultaneously belong to two or more emotional categories. As NLP technology continues to innovate and evolve, multi-label emotion datasets enriched with nuanced micro-emotions are expected to replace the initial, singular annotations of single-label datasets [11]. SemEval-2007 serves as a micro-emotion dataset utilizing news headlines as its corpus, annotated with emotion labels carrying effective values, yet it comprises a relatively small sample size of 1,250. In 2020, at the ACL conference, Google released the large, manually annotated GoEmotions multi-label emotion dataset [12].Despite being considered the largest fine-grained emotion dataset currently available [13], the GoEmotions dataset still faces issues of label sparsity and imbalance. Specifically, single-label samples account for 83%, while dual-label samples comprise only 15%, and samples with three or more labels make up a mere 2%.The sparsity of labels may hinder model learning [14], particularly due to imbalances where some labels appear infrequently, resulting in poor predictive performance, while others occur frequently, leading to biased predictions favoring those labels.

In recent years, with the rapid development of human-machine alignment [15] and humanoid robots [16], there has been an increasing demand for machines to understand human emotions, making affective computing [17] and micro-expressions [18,19] prominent research topics. While macro-expressions provide relatively straightforward and direct representations of emotions, micro-expressions more accurately reflect subtle, unconscious, or fleeting emotional states. Annotating and detecting micro-expressions presents a greater challenge. Current methodologies for capturing and detecting micro-expressions through images or videos have resulted in the development of several datasets with emotional intensity related to micro-expressions [20–24], alongside published research findings. Compared to micro-expressions, the subtle micro-emotions embedded in natural language expressions can more comprehensively capture the nuanced emotional fluctuations present in human language and text. Achieving human-machine alignment in humanoid robots and human-machine dialogue necessitates that the recognition and detection of micro-emotions in textual communication be regarded as equally important as that of micro-expressions. Currently, the scarcity of publicly available datasets for text-based micro-emotions poses a challenge for research in this domain.

In the context of natural language, micro-emotions refer to fleeting, low-intensity, and often subconscious emotional states expressed subtly through text. These emotions are typically harder to detect than macro-emotions such as joy or anger, as they are conveyed through nuanced wording, implicit sentiment, or slight linguistic variations. For instance, a sentence like "That's exactly the kind of brilliant nonsense I expected" may carry sarcasm mixed with disappointment and helplessness—emotional shades that would be missed by coarse labeling systems. Micro-emotions provide a deeper view into the speaker's internal psychological state, making them highly valuable in fields such as sentiment analysis, mental health screening, and user experience optimization. Our study considers micro-emotions as integral components of fine-grained emotion modeling and aims to capture them systematically through quantification and annotation.

The capacity of machines to understand textemotion has been a long-term research objective in natural language processing (NLP). Currently, emotion datasets—whether annotated for single-label, multi-label, or micro-expressions—primarily rely on manual annotation. This reliance is inevitably influenced by external and subjective factors, resulting in high costs, low efficiency, and challenges in annotating micro-emotions. To better capture the various subtle nuances of human emotions in emotion detection, exploring machine or machine-assisted micro-emotion annotation datasets has become increasingly vital. We have sought to establish a simple yet effective framework for micro-emotion detection and annotation, termed the Expansion Quantization Network (EQN), which incorporates energy scores. Within the EQN framework, the automatic multi-label annotation assigns the highest energy values to macro-emotions and lower energy values to micro-emotions, thereby enhancing the model's ability to understand and predict emotional nuances more effectively.Our EQN framework is adaptable to manually annotated single-label or multi-label emotion datasets.

**Contributions of this paper**:

1. **Introduction of continuous emotional intensity**: The EQN framework adds continuous energy values to samples based on manually annotated single-label or multi-label emotion datasets. By quantifying emotional intensity with continuous values, the framework distinguishes between macro-emotions and micro-emotions, addressing the subjectivity inherent in manual annotations.

2. **Full label mapping numerical method**: This method learns the interdependencies among data labels to annotate label values without the need for prior knowledge or emotional lexicons, thereby reducing the risk of data contamination.

3. **Label regression method for training sets**: By learning from the fully annotated training set, this approach regresses the labels that have already been manually annotated to a maximum value while retaining the values of automatically annotated labels. This method enhances the performance of training iterations.

4. **Validation of the EQN framework's generalization ability in NLP models**: Comparative experiments conducted on five distinct single-label and multi-label emotion detection datasets using various NLP models demonstrate that the EQN framework is widely applicable across NLP models.

5. **Supplementary annotation of the GoEmotionsmicro-emotion dataset and public release:**GoEmotions is a 28-class emotion dataset, which presents significant challenges for emotion classification and micro-emotion detection. Utilizing the EQN framework, we first fully annotated the GoEmotions dataset and applied the proposed label regression method to supplement the micro-emotion annotations, which have now been publicly released.

## Related work

Machine-assisted annotation of micro-emotion labels with energy values is particularly crucial in applications such as customer emotion management, psychological health analysis, and brand monitoring. It provides more nuanced emotional feedback and has garnered significant attention from scholars in the field of Natural Language Processing (NLP). The SemEval-2007 dataset [11], which includes effective value annotations, is a multi-label micro-emotion dataset based on

manual annotation. Although it offers micro-emotion data for machine-assisted labeling, its size is relatively limited. Early research in emotion focused primarily on emotional polarity (such as positive, negative, and neutral), typically employing bag-of-words models or emotion dictionaries for classification [1]. Each emotional lexicon in these dictionaries is assigned a score to evaluate the accuracy of its emotional sentiment. S. Saifullah et al. [25] employed machine learning methods, utilizing data that underwent preprocessing through tagging, filtering, stemming, tokenization, and emoji conversion. By leveraging 24 combinations of ML and FE algorithms, they achieved optimal performance in anxiety emotion detection. The Chinese EmoBank [26] provides a manually annotated Chinese dimensional emotion lexicon, which includes various modal words to express emotional intensity. Additionally, research has proposed methods to generate word-level emotional distribution (WED) vectors by integrating domain knowledge with dimensional dictionaries [27]. The latest study by S. Saifullah et al. [28] employed semi-supervised learning techniques for automated annotation, achieving remarkable results in hate speech detection. In semi-supervised learning, their model learns from labeled data, which provides explicit information, while also extracting implicit knowledge from unlabeled data. This hybrid approach enables the model to generalize effectively and make informed predictions even when labeled data is limited. Moreover, this method enhances the model's ability to handle real-world scenarios where annotated data is scarce.In 2024, the latest research by Wang Yaoqi [29] and colleagues attempted to introduce emotional distance among emotions, utilizing a text EDLE method that incorporates VAD emotional knowledge to enhance label accuracy based on emotion dictionaries.

Despite the presence of energy intensity scores in emotion dictionaries, these resources are primarily utilized for determining the categorical attributes of emotions and do not yet facilitate the automatic annotation of emotional energy intensity values. In multi-label learning (MLL) methods, the objective is to identify multiple emotions for each sentence [30]. This approach involves setting a threshold, whereby emotions scoring above this threshold are marked as relevant, while others are deemed irrelevant. However, MLL methods are ineffective in learning the intensity of each individual emotion.

To address this issue, Geng (2016) [31] proposed a novel machine learning paradigm known as label distribution learning (LDL). Subsequently, the emotional distribution learning (EDL) algorithm improved upon the label distribution framework [32]. However, these methods necessitate the design of complex textual features, which require substantial human resources. In 2024, EmoLLMs [33], based on large language models such as ChatGPT, employed instruction data to fine-tune various LLMs with the aim of predicting both the emotional category and intensity of the input text. EmoLLMs are capable of generating micro-emotion labels accompanied by numerical values. Although this approach has demonstrated promising results, the process of developing instruction tuning data remains intricate, with the resultant emotional classification and intensity largely contingent upon the cognitive capabilities of the large models.

In summary, scholars in the field of NLP have been dedicated to uncovering the emotional energy values embedded in text, employing a variety of distinctive methods. However, these approaches still exhibit limitations in practical applications, failing to achieve the automatic annotation of large-scale micro-emotion datasets. Our EQN framework is capable of automatically annotating labels with continuous values, enabling multi-label datasets to encompass both macro and micro emotional characteristics. The principles and methods underlying this framework are relatively straightforward, yet its applicability is broad.

Micro-emotion detection is one of the crucial downstream tasks for multi-label micro-emotion datasets. Traditional machine learning models typically classify text by converting it into word vectors and extracting features through feature engineering, with commonly employed methods including Naive Bayes [34], Support Vector Machines (SVM) [35], and Logistic Regression [36]. In contrast, deep learning models leverage neural networks to autonomously learn hierarchical feature representations from data, extracting rich and complex features from the raw input. This process often necessitates substantial amounts of training data. For instance, Convolutional Neural Networks (CNN [37]) and Recurrent Neural Networks (RNN) [38], including Long Short-Term Memory (LSTM) networks and Gated Recurrent Units (GRU), are effective in extracting textual features and performing classification. Currently, fine-tuning methods based on large language models, such as BERT and GPT, are widely applied to multi-label micro-emotion datasets, demonstrating favorable results.

In the section dedicated to validating the EQN framework, we conducted comparative experiments using five models, including Artificial Neural Networks (ANN), Deep Convolutional Networks (CNN), Recurrent Neural Networks (RNN), TextCNN, and BERT. In these experiments, all models employed default parameters without any specialized tuning, with the primary objective of assessing the usability and generalization capabilities of the EQN framework. During the process of annotating a large micro-emotion dataset using the EQN framework, we supplemented the GoEmotions micro-emotion dataset with additional annotations based on the BERT model, and we performed various evaluations of the annotation results, simultaneously comparing them with relevant literature.

The main components of this paper are as follows: The first and second sections present the introduction and related work; the third section outlines the fundamental structure and operational mechanisms of the EQN; the fourth section conducts comparative experiments on five sets of single-label and multi-label emotion datasets, examining the differences between using and not using the EQN framework to validate its generalization capabilities through traditional neural networks, deep learning, and large language models; the fifth section employs the complete EQN framework to experiment with the GoEmotions dataset, which encompasses 28 categories of emotions, and compares the results with evaluation metrics from relevant literature [13], thereby further validating the effectiveness of the EQN framework; finally, the paper concludes with a summary of findings.

## Methods

This section provides a comprehensive overview of the EQN, including a flowchart depicting the overall process of the EQN framework, definitions of key terms, the operational steps of the EQN framework, the core structure, as well as the design of both the input and output components.

### EQN framework

The process of using machine models to detect data samples generally comprises three components: data input, model learning and processing, and classification output. The EQN proposed in this paper primarily focuses on enhancing the input and output components, making it compatible with any machine learning model designed for NLP classification tasks. The structure of the EQN framework is illustrated in Fig 1.

**Terms involved in the EQN.  Full label:** Refers to the data samples that are initialized or output with complete category labels, each accompanied by a real number representing emotional intensity. In this paper, the range of real values for full labels is specified to be between 0.0 and 10.0. For samples lacking corresponding predefined label attributes, the minimum emotional intensity is marked as 0.0, while the maximum emotional intensity is set at 10.0. After automatic annotation, thresholds can be adjusted according to actual conditions.

**Full label initialization:** This process maps the manually annotated single or multi-labels from the original dataset to values of 0.0 or 10.0 prior to the initial run of the training set. Labels that have been manually annotated are assigned the maximum value of 10.0, whereas unannotated labels receive the minimum value of 0.0.

**Training set label regression:** Based on the core framework of EQN, this step involves annotating the training set with full labels, assigning the maximum value of 10.0 to the labels that have been manually annotated, while other values remain unchanged. In other words, it replaces the 0.0 values used during the initialization of full labels with the micro-emotional values learned by the model.

**Operational process of the EQN framework.**  The EQN framework consists of a two-stage training pipeline for enhancing emotion classification through full-label regression. Its operation is straightforward and can be summarized as follows:

1. **Data Preparation**: Each training sample with a single emotion label is transformed into a multi-dimensional one-hot vector representing the full emotion label space (e.g., for 28 emotions in GoEmotions).

**Fig 1. Schematic diagram of the EQN structure.**

2. **Stage 1 – Full Label Initialization & Model 1 Training**:

A base classification model (Model 1) is trained on the initialized dataset using standard loss functions (e.g., MSE loss for regression).

Model 1 learns to map input texts to emotion label vectors.

3. **Soft Label Generation**:

Model 1 is used to predict probability distributions over the entire label space for each training sample.

These predicted probabilities serve as **soft labels** for all emotions not originally annotated (while preserving the original label as 1.0).

4. **Stage 2 – Model 2 Training with Refined Labels**:

A second model (Model 2) is trained on the soft-labeled dataset to learn a more robust and generalized representation of emotional features.

5. **Inference**:

Model 2 can be used to classify emotions of new texts or assign multi-dimensional emotional scores for downstream tasks.

Note: Steps 1–3 are referred to as the **core EQN module (CoEQN)**, which is also used as a standalone component in Section 4 for ablation experiments.

A full Python-style pseudocode is provided in the Supplementary Material to ensure reproducibility.

Based on the aforementioned EQN framework model diagram, the primary distinction between the EQN framework and conventional text classification models lies in the input and output components. The following sections will focus on detailing the input and output components of the EQN framework.

**EQN input component. Typical text input:** In traditional NLP tasks for single-label or multi-label text classification, the input text undergoes preprocessing and feature extraction (Tokenization, Embedding) before being converted into numerical features to serve as input. The input component generally comprises the text along with its corresponding labels. Labels are typically represented as either integers or one-hot encodings and possess the following characteristics:

1. **Integer label representation**: For the i-th sample, the input feature $X_i$ corresponds to the label category $Y_i$, with N categories represented as 0, 1, 2,...,N − 1, where N denotes the total number of categories.

2. **One-Hot encoding of labels:** Each label is treated as an independent feature and represented as a binary encoding, with only two possible values: 0 and 1. Here, 0 indicates the absence of the emotional label in the sample, while 1 signifies the presence of that emotional label.

**Input for the EQN framework:** In the EQN framework, the processing of input text data aligns with traditional methods; however, the representation of labels differs significantly. In conventional approaches, whether using integer or one-hot encoding, labels merely indicate the presence or absence of a category. In contrast, the labels for input text in the EQN framework are annotated as full label values.

The input method for full label values has the following characteristics:

1. **Full category label annotation**: It employs full category labels, with each label assigned a continuous real number representing the intensity of the emotion.

2. **Value range for labels:** The numerical range corresponding to the labels can be defined according to task requirements. In this study, the values are set between 0.0 and 10.0, where 0.0 signifies the minimum emotional intensity for the label, and 10.0 denotes the maximum intensity.

3. **Two input methods:** The initialization of full category label values for samples and the label regression of the training set correspond to two distinct frameworks: CoEQN and EQN, respectively.

As shown in Fig 1, full-label numerical initialization mapping and label regression pertain to the input component. In the CoEQN framework, the full-label numerical initialization mapping assigns an initial value to each emotion label, using real numbers between 0 and 10 to represent the intensity of emotions. For each label $t_j$ in sample i, initialization mapping can be performed using the following formula.

$$Y_{ij} = f(t_j) \tag{1}$$

Here, $f(t_j)$ is a mapping function that assigns the value mapped from label $t_j$ of sample i to $Y_{ij}$.

$$f(t_j) = \begin{cases} 10.0, & \text{if sample i has label } t_j \\ 0.0, & \text{if sample i does not have label } t_j \end{cases} \tag{2}$$

Label regression in the EQN framework involves using the CoEQN-trained model to annotate the training set, followed by performing label regression on the annotated training set. Let E represent the intensity value annotated by the model; the label regression formula is as follows:

$$f(t_j) = \begin{cases} E = 10.0, & \text{if sample i has label } t_j \\ E = E, & \text{if sample i does not have label } t_j \end{cases} \tag{3}$$

EQN is a framework that takes full-label text input, processes it through model learning, and outputs the intensity value of each label via linear regression, providing full-category label intensity values for each predicted sample. By individually training a linear regression model for each label, the framework generates an intensity prediction for each label. This intensity value, a continuous measure (ranging from 0 to 10 in this study), reflects the relevance or association between the label and the current text. By setting an intensity threshold, the framework determines which labels are present, thereby enabling macro-emotion and micro-emotion annotation. Emotion classification is achieved by ranking the labels based on the annotated intensity values.

**Examples of traditional input and EQN framework input:** To clarify the differences between the full label method and the integer or one-hot encoding labeling approaches, the following comparative examples are presented.

Assuming the dataset *X* contains three samples—Sample 1, Sample 2, and Sample 3—with three predefined labels labeled as 0, 1, and 2. The manual annotation results indicate that Sample 1 has labels 0 and 1, Sample 2 has label 1, and Sample 3 has label 2. The input data for Samples 1, 2, and 3 can be represented using full label initialization, full label regression, integer encoding, and one-hot encoding, as shown in Table 1.

**EQN output component.** The design of the output component in the EQN framework differs from that of traditional multi-label classification due to the distinct problems it addresses. In conventional multi-label classification, input text is assigned to predefined categories, with the initialized label values being discrete and the output resulting in discrete categories.

While the results produced by the EQN framework can be utilized for classification tasks, it primarily addresses regression tasks, yielding continuous values. This approach is akin to systems used for stock price prediction or real estate valuation, where the output is expressed as numerical values.

In the EQN framework designed to solve regression problems, the model's output component first connects to a fully connected Dense layer. Each neuron in the Dense layer is linked to all neurons in the previous layer, with each connection assigned a weight that learns the relationships between different features, thereby preparing data for subsequent output.

**Table 1. Examples of full Label method and labels represented by integers or one-hot encoding.**

| X | Labels | One-Hot Encoding | Integer | Full Label Initialization | Full Label Regression |
|---|---|---|---|---|---|
| Sample 1 | 0, 1 | [1,1,0] | 0, 1 | [10.0,10.0,0.0] | [10.0,10.0,5.2] |
| Sample 2 | 1 | [0,1,0] | 1 | [0.0,10.0,0.0] | [0.0,10.0,4.0] |
| Sample 3 | 2 | [0,0,1] | 2 | [0.0,0.0,10.0] | [8.5,6.1,10.0] |

**Notes:**

·In the "Full Label Initialization" and "Full Label Regression" columns, the values represent the intensity of emotions associated with each label.

·The "Integer Encoding" column indicates the presence (1) or absence (0) of labels.

·The "One-Hot Encoding" column shows the binary representation of labels, where only one position is marked as 1, indicating the presence of a specific label.

The final layer of the network is a linear layer (using a linear activation function) that consists of C units, whereCrepresents the total number of categories. Corresponding to the full label input of samples in the EQN framework, the output section of the linear layer produces Cchannels, each outputting the intensity score of the sample for each label, thereby achieving full label output of emotional values.

The output of the EQN framework provides specific numerical values, which not only address regression problems but can also be utilized to solve sample classification issues using the annotated values. Each of the C channels in the linear layer outputs the intensity score for each label, with scores ranging from 0.0 to 10.0. Here, 0.0 indicates the absence of the corresponding emotional label, while 10.0 signifies a very strong presence of that emotional label. Values between 0.0 and 10.0 represent varying levels of emotional intensity.

By setting a threshold, multi-label classification can be performed (as demonstrated in the fourth part of this paper, where annotated data is used for classification, serving as one method to validate the annotation effectiveness of the EQN framework).

Assume the dataset contains $m$ samples. For each text with an n-dimensional vector, the feature vector is represented as $x = [x_1, x_2, \ldots, x_n]$. These features may include term frequency, TF-IDF, word embeddings, etc. With N labels, the model's objective is to predict an intensity value $\hat{y}_j$ for each label j ($j = 1, 2, \ldots, N$). The model outputs an energy level score for each label, forming an energy level prediction vector $\hat{y}_{ij}$ for the j-th label of the i-th sample, corresponding to the true value $y_{ij}$.

$$\hat{y}_{ij} = [\hat{y}_{i1}, \hat{y}_{i2}, \ldots, \hat{y}_{iN}] \tag{4}$$

$$y_{ij} = [y_{i1}, y_{i2}, \ldots, y_{iN}] \tag{5}$$

Here, $y_{ij} = 10$ indicates that the true value for the j-th label is present, while $y_{ij} = 0$ indicates that the true value for the j-th label is absent. The intensity prediction value for the j-th label is estimated via linear regression as follows:

$$\hat{y}_{ij} = w_{ij}^T x + b_{ij} \tag{6}$$

Here, $w_{ij}$ represents the weight vector corresponding to label j for text i, and $b_{ij}$ is the bias term. When selecting the optimal model, the EQN framework employs the Mean Squared Error (MSE) as the loss function to measure the difference between predicted values and true label values. By minimizing the gap between predicted and true labels, the framework optimizes the weights $w_{ij}$ and bias $b_{ij}$. The loss function $L(w_{ij}, b_{ij})$ is calculated as follows:

$$L\left(w_{ij}, b_{ij}\right) = \frac{1}{2m} \sum_{i=1}^{m} \left(\hat{y}_{ij} - y_{ij}\right)^2 \tag{7}$$

After the model has been trained, the final predicted value $\hat{y}_{ij}$ is selected as the output for the EQN framework, which provides full-category labels with emotional intensity. For emotion annotation, a threshold h is set based on the specific context. Labels with values below this threshold are considered to indicate the absence of that particular emotion, and their value is set to 0. The specific annotation formula is as follows:

$$f(t_j) = \begin{cases} \hat{y}_{ij} & \text{if } \hat{y}_{ij} \geq h \\ 0 & \text{if } \hat{y}_{ij} < h \end{cases} \tag{8}$$

**EQN framework evaluation metrics.** Statistical classification in emotion detection is a crucial downstream task. Key evaluation metrics in dataset classification include Precision Recall and F1-score etc. For single-label classification in emotion datasets, the full-category output for each sample is processed using Max(), where the label with the highest energy level is selected as the predicted label, which is relatively straightforward. In multi-label classification, since each sample may have multiple labels, the computation of evaluation metrics becomes more complex.

For emotion classification, in the case of single-label classification, the full-category output for each sample is processed using Max(), and the label with the highest energy level is chosen as the predicted label. In multi-label classification, the number of labels for the sample and the threshold are used to select the predicted labels. Assuming sample i has $k_i$ true labels, we select the top $k_i$ labels with the highest energy scores as the predicted labels. The predicted label set is as follows:

$$\hat{T}_i = \text{TOP\_k}_i\left(\hat{y}_{ij}\right) \tag{9}$$

The following are the calculation formulas for the evaluation metrics of the EQN framework. These formulas compute the label matching ratio at the sample level, and the overall evaluation metric is obtained by averaging across all samples.

Precision measures how many of the predicted labels are correct. The precision for the i-th sample is calculated as:

$$precision_i = \frac{|T_i \cap \hat{T}_i|}{|\hat{T}_i|} \tag{10}$$

Where $T_i$ is the true label set of the i-th sample, and $\hat{T}_i$ is the label set predicted based on the energy scores. $|T_i \cap \hat{T}_i|$ represents the intersection of the true label set and the predicted label set, i.e., the number of correctly predicted labels.

The overall multi-label precision of the EQN framework is the average of the precision values for all samples. The calculation formula is as follows:

$$precision = \frac{1}{m} \sum_{i=1}^{m} \frac{|T_i \cap \hat{T}_i|}{|\hat{T}_i|} \tag{11}$$

Recall measures how many of the true labels are correctly predicted. The recall for the $i$-th sample is calculated as follows:

$$Recall_i = \frac{|T_i \cap \hat{T}_i|}{|T_i|} \tag{12}$$

The overall multi-label recall of the EQN framework is the average of the recall values for all samples. The calculation formula is as follows:

$$Recall = \frac{1}{m} \sum_{i=1}^{m} \frac{|T_i \cap \hat{T}_i|}{|T_i|}$$

(13)

The F1-score is the harmonic mean of precision and recall, balancing both metrics. The F1-score for the i-th sample is calculated as follows:

$$F1_i = \frac{2 \times \text{precision}_i \times \text{Recall}_i}{\text{precision}_i + \text{Recall}_i}$$

(14)

The overall F1-score of the EQN framework is the average of the F1-scores for all samples. The calculation formula is as follows:

$$F1 = \frac{1}{m} \sum_{i=1}^{m} \frac{2 \times \text{precision}_i \times \text{Recall}_i}{\text{precision}_i + \text{Recall}_i}$$

(15)

The EQN framework comprises two processes: CoEQN and EQN, with the latter encompassing the entire workflow for the automatic annotation of emotional datasets and micro-emotion detection. CoEQN includes only step1–5 of the EQN framework.

To validate the applicability of the EQN framework, experiments were conducted using the same processes and parameter settings as conventional models. This approach enhances the comparability of the experimental results and also demonstrates the generalizability of the EQN framework.

## Experiments

To validate the broad applicability of the EQN framework, we selected five commonly used datasets (four single-label and one multi-label), along with various algorithms and language models for comparative analysis. The experimental setup, including the equipment specifications, experimental rules, and model evaluation methods, will be detailed.

The results of the tests comparing "with" and "without" the CoEQN framework will be presented in different formats, including tables and Pearson correlation coefficient heatmaps. These visual representations will highlight the outcomes of the same model under identical conditions, thereby validating the usability of the EQN framework.

### Datasets used for comparative experiments

**7health dataset.** The 7health dataset [4] is a mental health emotion analysis dataset designed to reveal psychological health patterns through statements. This comprehensive dataset is a meticulously curated collection of mental health statuses tagged from various statements. It amalgamates raw data from multiple sources, which have been cleaned and compiled to create a robust resource for developing chatbots and conducting emotion analysis.

The dataset comprises 51,074 entries, annotated with seven categories of emotions: anxiety, bipolar, depression, normal, personality disorder (PD), stress, and suicidal. The distribution of sample counts for each category is presented in Table 2.

As indicated by the data in Table 2, the sample counts in this dataset are severely imbalanced. With the exception of the normal category, the other six categories represent negative emotions that exhibit significant similarity, making it a particularly challenging multi-label dataset.

 

**Table 2. Distribution of sample counts in the 7health dataset.**

| class | anxiety | bipolar | depression | normal | PD | stress | suicidal |
|-------|---------|---------|------------|--------|------|--------|----------|
| count | 3888 | 2877 | 15404 | 16351 | 1201 | 2669 | 10653 |

**6emotions dataset.** The 6emotions dataset [5] is an English corpus comprising six categories of emotions. Each entry in this dataset consists of a text segment representing a Twitter message, along with a corresponding label that indicates the predominant emotion conveyed. The emotions are classified into six categories: sadness (0), joy (1), love (2), anger (3), fear (4), and surprise (5). This dataset provides a rich foundation for exploring the nuanced emotional landscape within the realm of social media.

**3TFN dataset.** The Twitter Financial News dataset(3TFN dataset) [6] is an English-language dataset containing an annotated corpus of finance-related tweets. This dataset is used to classify finance-related tweets for their emotions.The dataset holds 11,932 documents annotated with 3 labels:Bearish, Bullish, Neutral。

**3TSA dataset.** The Twitter Sentiment Analysis Dataset (3TSA dataset) [7] is a three-class dataset comprising approximately 163,000 tweets, each associated with sentiment labels. The dataset consists of two columns: the first column contains the cleaned tweets and comments, while the second column indicates the corresponding sentiment label.

All tweets have been cleaned using Python's regular expressions and natural language processing techniques, with sentiment labels ranging from −1–1. A label of 0 indicates a neutral tweet, 1 denotes a positive sentiment, and −1 signifies a negative tweet.

**GoEmotions dataset.** At the 2020 ACL conference, researchers from Google released the GoEmotionsdataset [12], the largest and most finely grained multi-label micro-emotion dataset to date, comprising 58,000 manually annotated Reddit comments. This dataset expands the emotion categories to 28, providing an opportunity to better uncover users' latent emotions.

The dataset is divided into three parts: the training dataset contains 43,410 samples, the test dataset includes 5,427 samples, and the validation dataset comprises 5,426 samples. The emotion categories are as follows: admiration, amusement, anger, annoyance, approval, caring, confusion, curiosity, desire, disappointment, disapproval, disgust, embarrassment, excitement, fear, gratitude, grief, joy, love, nervousness, optimism, pride, realization, relief, remorse, sadness, and surprise.

## Models and configurations

This study employs five models—ANN [39], CNN [40], LSTM [41], TextCNN [1], and BERT [42]—for comparative testing across the selected datasets.

**ANN model.** Artificial Neural Networks (ANN) are computational models that mimic biological neural systems and are widely applied in machine learning and artificial intelligence. ANN serves as the foundation of modern deep learning, with many complex models (such as Convolutional Neural Networks and Recurrent Neural Networks) developed based on this fundamental structure. It primarily comprises an input layer, hidden layers, and an output layer. In this study, the parameters for the ANN model are set as follows: the number of neurons in the input layer is 512, the number of neurons in the hidden layer is 256, and the activation function is ReLU.

**CNN model.** Convolutional Neural Networks (CNN) are extensively used in text classification due to their effectiveness in capturing local features and contextual information. A typical CNN architecture includes an input layer, convolutional layers, pooling layers, dense layers, and an output layer. For the CNN model utilized in this study, the parameters are set as follows: `max_features = 15000`, the number of input channels is 32, the number of convolutional filters is 7, and the activation function is ReLU.

**LSTM model.** Long Short-Term Memory networks (LSTMs) are a specialized type of Recurrent Neural Network (RNN) capable of learning long-term dependencies. By incorporating a complex internal structure with multiple gating

mechanisms, LSTMs effectively regulate the flow of information, allowing the network to retain long-term memories when necessary and discard irrelevant information when it is no longer needed. In this study, the LSTM module provided by TensorFlow is employed directly. The parameters for the LSTM model are set as follows: `input_dim = 5000`, `output_dim = 150`, `input_length = 150`, and the hidden layer size of the LSTM layer is 128.

**TextCNNmodel.** TextCNN is a convolutional neural network model specifically designed for text classification, improving upon standard CNN architectures by modifying the convolutional layers. TextCNN employs convolutional layers with filters of varying sizes, where each filter is responsible for extracting specific n-gram features. Different-sized filters (e.g., 1-gram, 2-gram, 3-gram) capture contextual information of varying lengths. The parameters for the TextCNN model in this study include three 1D convolutional layers, each with filter sizes of 3, 4, and 5, and a channel size of 256.

**BERT model.** BERT excels in emotion detectiondue to its robust contextual understanding and flexible training strategies, making it highly effective for emotion detection tasks [43]. It processes both left and right contexts in sentences, allowing the model to comprehend word meanings and contexts more accurately. This bidirectional processing is crucial for emotion detection, as words may carry different emotions based on contextual variations. We implement fine-tuning of the BERT-base-cased pre-trained model for text classification tasks in our study.

## Experimental environment and rules

The experimental platform and key parameters for this study are as follows:

**GPU**: NVIDIA GeForce RTX 3090 GPU;

**BERT model runtime environment**:python=3.7, pytorch=1.9.0, cudatoolkit=11.3.1, cudnn=8.9.7.29;

**Other modelsruntime environment**: python=3.8, tensorflow==2.6.0, cudatoolkit=11.3.1, cudnn=8.2.1;

**Parameter settings**: The text length or sequence length is uniformly set to 150 for all models. For text input, except for BERT, TensorFlow's Tokenizer and sequence representations are consistently utilized. The text input for BERT employs a summation of three types of embeddings (Token, Segment, Position) to generate the final input representation for each word.

**Rules**: To ensure consistency with the operational workflow of the five comparative models, the CoEQN framework is employed for validation in this section. For the same model, in the comparative experiments of "using" versus "not using" the CoEQN framework, the fundamental structure, parameter settings, training set, and test set samples remain unchanged. In the experiments where the CoEQN framework is utilized, only the input and output portions are modified accordingly, while the parameters follow those detailed in Section "Experiments". The labels of the training set are mapped to full labels and initialized with energy level values, while the output for the test set generates full label energy scores. Classification is performed based on the full label energy scores of the test set using either MAX() (for single-label classification) or a predetermined threshold (for multi-label classification), with results compared to those obtained from conventional model methods on the same test set.

## Results and discussion

**Comparison of test results.** In this section, we present a detailed comparison of the results obtained from the different classification models—specifically focusing on single-label, multi-label, and EQN full label mapping (EQN) approaches. Based on the aforementioned rules, the testing results for five datasets and five models utilizing conventional single-label and multi-label classification methods, as well as the EQN full-label approach, are presented in Table 3.

The results indicate that, from traditional neural network models like ANN to deep learning models such as CNN, RNN, TextCNN, and BERT, the basic structure, parameter settings, training datasets, and test datasets remain consistent

**Table 3. Testingresults of single-label, multi-label, and full-label mapping classification models.**

| model/dataset | | 7health | 6emotions | 3TFN | 3TSA | 28GoEmotions |
|---|---|---|---|---|---|---|
| ANN | S | 0.3202 | 0.3358 | 0.6557 | 0.5199 | |
| | M | | | | | 0.1611 |
| | EQN | 0.374 | 0.3551 | 0.6482 | 0.5254 | 0.1877 |
| CNN | S | 0.6658 | 0.6778 | 0.7353 | 0.8166 | |
| | M | | | | | 0.3892 |
| | EQN | 0.6836 | 0.7113 | 0.7592 | 0.8161 | 0.4076 |
| TextCNN | S | 0.6624 | 0.7603 | 0.7575 | 0.7357 | |
| | M | | | | | 0.3889 |
| | EQN | 0.6672 | 0.7869 | 0.7717 | 0.7448 | 0.4205 |
| LSTMS | S | 0.7068 | 0.9318 | 0.8099 | 0.93 | |
| | M | | | | | 0.4545 |
| | EQN | 0.7295 | 0.9352 | 0.8073 | 0.9283 | 0.4658 |
| BERT | S | 0.7855 | 0.9328 | 0.8378 | 0.9405 | |
| | M | | | | | 0.4665 |
| | EQN | 0.8034 | 0.935 | 0.8458 | 0.9413 | 0.4849 |

**Table notation**: single-label is denoted as S, multi-label as M, and EQN full label mapping as EQN.

between the "using" and "not using" CoEQN frameworks. This suggests that EQN is highly compatible with various models.

The full label mapping method employed by CoEQN has enhanced model performance, demonstrating particularly significant improvements in accuracy on datasets with a larger number of categories. Although the performance gains are less pronounced on datasets with fewer categories, the fundamental accuracy of the models has not been compromised. The full label numerical mapping method utilized within the EQN framework effectively capitalizes on the interdependencies among labels, providing continuous numerical annotations that enhance the model's understanding of subtle features. This demonstrates the excellent generalization capabilities of the EQN framework, making it suitable for a wide range of NLP models.

**Pearson correlation coefficient heatmap.** To further assess and demonstrate the quality of the automatic labeling achieved by the CoEQN framework, we calculated the Pearson correlation coefficients among the labels based on the distribution of the full label scores we annotated on the test set. According to Pearson's theory, the Pearson correlation coefficient ranges from −1 to +1; the greater the absolute value of the coefficient, the higher the degree of correlation. A negative value indicates an inverse correlation, while a positive value signifies a direct correlation.

For clarity, we use the test results from the 7health dataset as an example. By employing the BERT-based CoEQN framework, we annotated the full label scores on the 7health test set, and based on these scores, we generated a Pearson correlation coefficient heatmap, as depicted in Fig 2.

The Pearson correlation heatmap for the 7health dataset reveals that the personality disorder category exhibits low correlations with the other six health indicators. This aligns with the definition of personality disorders, which are characterized by persistent behavioral patterns stemming from genetic, congenital, and adverse environmental factors during individual development.

Furthermore, the correlation between depression and Normal is notably negative, with a coefficient of −0.43, followed by Suicidal and Normal at −0.38. This indicates that both depression and suicidal ideation significantly impact an individual's health. Interestingly, the correlation between depression and suicidal ideation is merely −0.07, suggesting a minimal relationship, which may seem counterintuitive. Literature [44] suggests that suicidal thoughts are common in depression

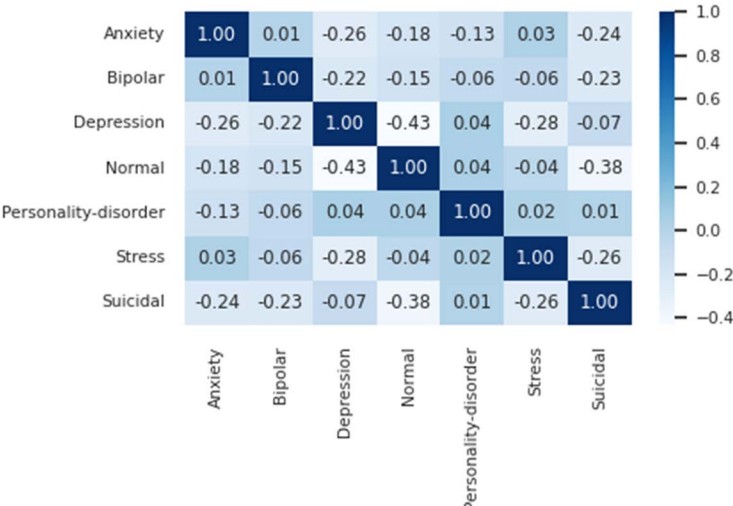

**Fig 2. Pearsoncorrelation heatmap for the 7health dataset.**

but are only moderately correlated with severe depression. Since the 7health dataset focuses on mental health rather than specific psychological disorders, the low correlation between depression and suicidal ideation is consistent with psychological understanding.

These evident correlations underscore the overall rationality of utilizing the EQN framework for emotional annotation, indirectly validating its practicality and applicability. It is hoped that these insights will provide a solid theoretical foundation for research conducted by mental health professionals and psychologists.

## Application: Annotation of the GoEmotionsdatasetbased on the EQN

GoEmotions is a fine-grained, multi-label emotion dataset characterized by a substantial manual annotation workload and significant classification challenges, providing valuable support for emotion detection. However, the labeling may be insufficient. To further evaluate the EQN framework's capability in capturing subtle emotions, we aim to supplement the annotation of the 28 categories within the GoEmotions dataset.

Utilizing the BERT model, we employ both the CoEQN framework and the EQN approach for automated annotation of the GoEmotions dataset. The analysis encompasses statistical evaluations of the annotation results, calculations of assessment metrics, and the generation of a Pearson correlation coefficient heatmap, which will be compared against the findings published in Google's dataset literature [13].

### GoEmotionsdata distribution

The distribution of sample labels in the training and testing sets of the GoEmotions dataset is illustrated in Fig 3. The training set comprises 43,410 samples across 28 categories, with the distribution of sample labels as follows: 36,308 samples are single-label, accounting for 84% of the total, while there are 28 samples with four labels and 1 sample with five labels. The l GoEmotions testing set contains 5,427 entries, with a maximum of 4 labels per entry. It includes 4,590 single-label samples, 774 samples with two labels, 61 samples with three labels, and 2 samples with four labels.

### CoEQN and EQN annotated datasets and comparative experiments

Section "Comparison of Test Results" details the detection experiments conducted on the GoEmotions dataset utilizing the CoEQN framework. This section presents the results of the comprehensive experiments based on the EQN framework,

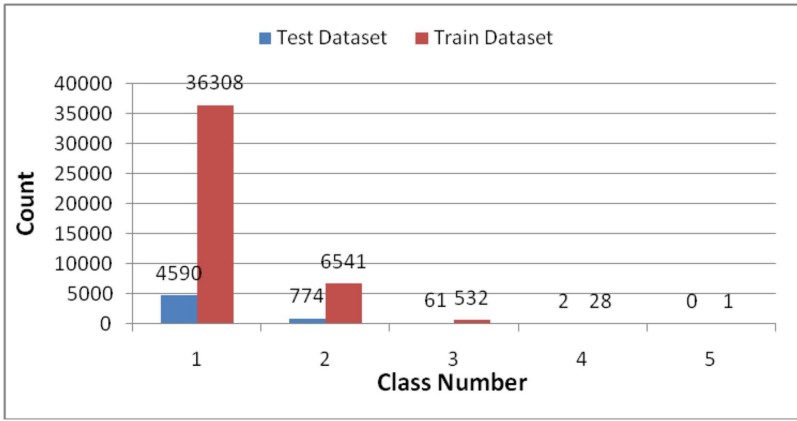

**Fig 3. Distribution of sample label counts in the GoEmotionstraining and testing sets.**

which also employs the basic BERT model with standard parameter settings, focusing on the automatic supplementary annotation of the GoEmotions dataset (both training and testing sets).

**Initialization of the training set and label regression of the training set.** Based on the CoEQN experiment, the full-label initialization method was employed to map the original GoEmotions training set, generating an initialized training set. Subsequently, the optimal BERT model was identified through learning to annotate both the GoEmotions training set and test set.

In the EQN experiment, building upon the CoEQN findings, the full-label regression method was applied to regress the manually annotated labels of the GoEmotions training set. After this regression, the test set was automatically annotated, and the results were analyzed and statistically presented.

The label regression training set method follows the principle of prioritizing manual annotations, where the selected model from CoEQN annotates the GoEmotions training set. The original manually annotated label values are restored to 10, while other learned micro-emotion intensity values remain unchanged. The segments for the initialized training set and the label regression training set are illustrated in Figs 4 and 5.

**Annotation of the GoEmotionstest set and comparative analysis.** In Section "Comparison of Test Results", the GoEmotions test set was annotated based on the CoEQN framework, as depicted in Fig 6. Under the same BERT model and parameter settings, the GoEmotions test set was automatically annotated using the EQN framework and the label regression training set. The annotation segments for the GoEmotions test set based on CoEQN are illustrated in Fig 6, while the segments based on EQN are shown in Fig 7.

Based on the two annotated GoEmotions test sets mentioned above, the statistical analysis and comparison of results from the experiments are presented in Table 4.

The experimental results indicate that the EQN model, which is based on full-label regression for the GoEmotions training set, demonstrates a significant enhancement in recognition performance. This finding substantiates that the full-label numerical method employed in this study allows the model to effectively learn more nuanced textual features, thereby improving the quality of the manually annotated dataset.

To further observe the framework's ability to capture subtle emotions, we use the prediction results from the test set based on Table 4 as an example. For simplicity, we only compare the single-label samples, which make up the largest proportion of the test set, totaling 4,590 samples. The top three predicted intensity values, TOP1, TOP2, and TOP3, are extracted from the predicted energy values across the 28 categories for each sample. The corresponding prediction hit counts are presented in Table 5.

**Fig 4. Segments of the initialized training set in the CoEQNframework.**

| | text | labels | admiration | amusement | anger | annoyance | approval | caring | confusion | curiosity | desire | disappoi | disa |
|---|---|---|---|---|---|---|---|---|---|---|---|---|---|
| 1 | text | labels | admiratio | amusement | anger | annoyance | approval | caring | confusior | curiosity | desire | disappoir | disa |
| 2 | My favour | 27 | 0 | 0 | 0 | 0 | 0 | 0 | 0 | 0 | 0 | 0 | |
| 3 | Now if he | 27 | 0 | 0 | 0 | 0 | 0 | 0 | 0 | 0 | 0 | 0 | |
| 4 | WHY THE F | 2 | 0 | 0 | 10 | 0 | 0 | 0 | 0 | 0 | 0 | 0 | |
| 5 | To make h | 14 | 0 | 0 | 0 | 0 | 0 | 0 | 0 | 0 | 0 | 0 | |
| 6 | Dirty Sou | 3 | 0 | 0 | 0 | 10 | 0 | 0 | 0 | 0 | 0 | 0 | |
| 7 | OmG pEyTc | 26 | 0 | 0 | 0 | 0 | 0 | 0 | 0 | 0 | 0 | 0 | |
| 8 | Yes I hea | 15 | 0 | 0 | 0 | 0 | 0 | 0 | 0 | 0 | 0 | 0 | |
| 9 | We need m | 8, 20 | 0 | 0 | 0 | 0 | 0 | 0 | 0 | 0 | 10 | 0 | |
| 10 | Damn yout | 0 | 10 | 0 | 0 | 0 | 0 | 0 | 0 | 0 | 0 | 0 | |
| 11 | It might | 27 | 0 | 0 | 0 | 0 | 0 | 0 | 0 | 0 | 0 | 0 | |
| 12 | Demograph | 6 | 0 | 0 | 0 | 0 | 0 | 0 | 10 | 0 | 0 | 0 | |
| 13 | Aww... sh | 1, 4 | 0 | 10 | 0 | 0 | 10 | 0 | 0 | 0 | 0 | 0 | |
| 14 | Hello eve | 27 | 0 | 0 | 0 | 0 | 0 | 0 | 0 | 0 | 0 | 0 | |
| 15 | R/sleeptr | 5 | 0 | 0 | 0 | 0 | 0 | 10 | 0 | 0 | 0 | 0 | |
| 16 | [NAME] — | 3 | 0 | 0 | 0 | 10 | 0 | 0 | 0 | 0 | 0 | 0 | |
| 17 | Shit, I g | 3, 12 | 0 | 0 | 0 | 10 | 0 | 0 | 0 | 0 | 0 | 0 | |
| 18 | Thank you | 15 | 0 | 0 | 0 | 0 | 0 | 0 | 0 | 0 | 0 | 0 | |
| 19 | Fucking c | 2 | 0 | 0 | 10 | 0 | 0 | 0 | 0 | 0 | 0 | 0 | |
| 20 | that is w | 27 | 0 | 0 | 0 | 0 | 0 | 0 | 0 | 0 | 0 | 0 | |
| 21 | Maybe tha | 6, 22 | 0 | 0 | 0 | 0 | 0 | 0 | 10 | 0 | 0 | 0 | |
| 22 | I never t | 6, 9, 27 | 0 | 0 | 0 | 0 | 0 | 0 | 10 | 0 | 0 | 10 | |

**Fig 5. Segments of the label regression training set in the EQN framework.**

| | text | labels | admiration | amusement | anger | annoyance | approval | caring | confusion | curiosity | d |
|---|---|---|---|---|---|---|---|---|---|---|---|
| 1 | text | labels | admiratio | amusement | anger | annoyance | approval | caring | confusior | curiosity | d |
| 2 | My favour | 27 | 1.15 | 0.22 | 0.01 | 0 | 0.54 | 0.36 | 0.08 | 0.23 | |
| 3 | Now if he | 27 | 0 | 0.52 | 0.72 | 1.02 | 1.18 | 0.24 | 0 | 0 | |
| 4 | WHY THE F | 2 | 0.31 | 1.35 | 10 | 1.57 | 0.48 | 0 | 0.14 | 0.44 | |
| 5 | To make h | 14 | 0 | 0 | 0.79 | 1.19 | 1.1 | 0.25 | 0.05 | 0.24 | |
| 6 | Dirty Sou | 3 | 0.3 | 0.24 | 0.6 | 10 | 0.96 | 0 | 0.05 | 0.31 | |
| 7 | OmG pEyTc | 26 | 0.27 | 0.43 | 0.36 | 0.5 | 0.92 | 0.11 | 0 | 0.35 | |
| 8 | Yes I hea | 15 | 0.64 | 0.33 | 0 | 0.22 | 0.13 | 0.53 | 0 | 0.05 | |
| 9 | We need m | 8, 20 | 3.18 | 0 | 0 | 0 | 2.32 | 0.96 | 0.25 | 0.2 | |
| 10 | Damn yout | 0 | 10 | 1.63 | 1.53 | 1.53 | 1.33 | 0.03 | 0 | 0.03 | |
| 11 | It might | 27 | 0 | 0 | 0 | 0.2 | 1.38 | 0.47 | 0.39 | 0.54 | |
| 12 | Demograph | 6 | 0 | 0.43 | 0 | 0.19 | 1.3 | 0.19 | 10 | 3.23 | |
| 13 | Aww... sh | 1, 4 | 0.67 | 10 | 0.62 | 0.71 | 10 | 0.11 | 0.59 | 0.51 | |
| 14 | Hello eve | 27 | 0 | 0 | 0 | 0.01 | 1.85 | 1.05 | 0.76 | 1.47 | |
| 15 | R/sleeptr | 5 | 0 | 0 | 0 | 0.12 | 1.72 | 10 | 0.59 | 1.03 | |
| 16 | [NAME] — | 3 | 0.3 | 0.86 | 2.5 | 10 | 1.32 | 0.08 | 0 | 0 | |
| 17 | Shit, I g | 3, 12 | 0 | 0.56 | 1.92 | 10 | 0.29 | 0.01 | 0.06 | 0.25 | |
| 18 | Thank you | 15 | 0.69 | 0.05 | 0 | 0.24 | 0 | 0.35 | 0 | 0.26 | |
| 19 | Fucking c | 2 | 0.28 | 0.58 | 10 | 2.04 | 0.71 | 0 | 0 | 0 | |
| 20 | that is w | 27 | 0.19 | 0 | 0.12 | 0.43 | 1.21 | 0.22 | 0.27 | 0.28 | |
| 21 | Maybe tha | 6, 22 | 2.01 | 0.55 | 0 | 0.21 | 1.51 | 0.06 | 10 | 1.16 | |
| 22 | I never t | 6, 9, 27 | 0.61 | 0.12 | 0 | 0.21 | 2.64 | 0.2 | 10 | 0.59 | |

The statistical results presented in Table 5 indicate that employing the full-label method on originally single-labeled instances and performing label regression for dual-label annotation resulted in a 17% increase in hit rate. Furthermore, if the dataset is augmented to include three labels, the hit rate improves by 25%. This outcome suggests that a single manually labeled instance may correspond to two, three, or even more related or similar emotions. It also demonstrates that

Fig 6 table:

| text | labels | admiration | amusement | anger | annoyance | approval | caring | confusion | curiosity | desire | disappoint | disapprov | disgust |
|---|---|---|---|---|---|---|---|---|---|---|---|---|---|
| I'm real | 25 | 0 | 0.92 | 1.07 | 0.99 | 0.48 | 0.35 | 0.36 | 0.29 | 0.41 | 0.94 | 0.57 | 0.53 |
| It's wond | 0 | 5.59 | 0.87 | 0.21 | 0.23 | 1.09 | 0.21 | 0.19 | 0.07 | 0 | 0.22 | 0.14 | 0.22 |
| Kings fan | 13 | 2.7 | 0.33 | 0 | 0 | 0.98 | 1.16 | 0.22 | 0.46 | 0.56 | 0 | 0 | 0 |
| I didn't | 15 | 0.69 | 0.17 | 0 | 0.32 | 0 | 0.47 | 0 | 0.14 | 0 | 0.02 | 0 | 0.06 |
| They got | 27 | 0 | 0 | 0.5 | 1.05 | 1.33 | 0.19 | 0 | 0 | 0.16 | 0.66 | 0.77 | 0.22 |
| Thank you | 15 | 0.89 | 0.08 | 0.02 | 0.32 | 0 | 0.4 | 0 | 0.24 | 0 | 0 | 0 | 0.12 |
| You're w | 15 | 0.67 | 0 | 0 | 0.14 | 0.57 | 0.7 | 0 | 0.2 | 0 | 0.12 | 0 | 0.02 |
| 100%! Cor | 15 | 4 | 0.63 | 0.23 | 0.6 | 0.58 | 0.39 | 0 | 0 | 0 | 0 | 0.1 | 0.09 |
| I'm sorr | 24 | 0 | 0.49 | 0.94 | 1.13 | 0.76 | 0.56 | 0.49 | 0.18 | 0.17 | 0.86 | 0.47 | 0.53 |
| Girlfrien | 25 | 0.67 | 0 | 1.3 | 1.75 | 1.02 | 0.17 | 0 | 0 | 0.31 | 1.39 | 0.76 | 1.18 |
| [NAME] ha | 3,10 | 0 | 0 | 0.18 | 0.46 | 1.54 | 0.3 | 0.13 | 0.05 | 0.11 | 0.26 | 0.8 | 0 |
| Lol! But | 1,18 | 1.41 | 2.63 | 0.54 | 0 | 0.18 | 0.21 | 0.08 | 0 | 0.36 | 0.13 | 0.03 | 0 |
| Translati | 8 | 0 | 0 | 0.13 | 0.13 | 0.16 | 0.48 | 0 | 0.98 | 1.61 | 0.88 | 0.13 | 0.4 |
| It's grea | 0,7 | 5.38 | 0.58 | 0 | 0 | 1.08 | 0.18 | 0.41 | 1.01 | 0.08 | 0 | 0 | 0.06 |
| I've also | 14 | 2.21 | 1.16 | 0.09 | 0.46 | 1.46 | 0.14 | 1.06 | 1.52 | 0.43 | 0.72 | 0.3 | 0.69 |
| I never w | 10 | 0 | 0.43 | 1.17 | 1.36 | 1.92 | 0.21 | 0.45 | 0 | 0.54 | 1.09 | 1.51 | 0.7 |
| The thoug | 14 | 0.14 | 0 | 1.62 | 1.95 | 1.62 | 0.32 | 0.07 | 0.02 | 0.24 | 1.09 | 1 | 1.19 |
| if the pa | 25,27 | 0 | 0 | 0.14 | 0.77 | 1.56 | 0.58 | 0.22 | 0.22 | 0.18 | 0.58 | 0.95 | 0.21 |
| Triggered | 1 | 0.68 | 4.14 | 0.51 | 0.73 | 0.3 | 0.22 | 0.47 | 0.28 | 0 | 0 | 0.04 | 0 |
| I'm autis | 15 | 1.29 | 0.04 | 0.02 | 0.28 | 0.16 | 0.43 | 0 | 0.15 | 0 | 0.03 | 0 | 0.11 |
| I know yo | 1 | 0.15 | 0.88 | 1.97 | 1.83 | 2.2 | 0.51 | 0.7 | 0.32 | 0.03 | 0.52 | 1.01 | 0.52 |
| May regre | 24 | 0 | 0.19 | 0.18 | 0.38 | 0.95 | 0.33 | 1.92 | 3.23 | 0.14 | 0.42 | 0.71 | 0.18 |
| After he | 27 | 0 | 0.26 | 0.15 | 0.41 | 0.88 | 0.11 | 0.47 | 0.97 | 0.21 | 0.27 | 0.6 | 0.09 |
| Well, the | 27 | 0.26 | 0 | 0.08 | 0.35 | 1.17 | 0.23 | 0.17 | 0.21 | 0.06 | 0.16 | 0.55 | 0 |
| Watch Veg | 6,27 | 0 | 0.22 | 0.05 | 0.38 | 1.56 | 0.34 | 0.69 | 0.74 | 0.12 | 0.31 | 0.76 | 0.04 |
| Again, ov | 27 | 0.02 | 0 | 0.07 | 0.34 | 1.93 | 0.24 | 0.4 | 0.31 | 0.12 | 0.27 | 0.87 | 0 |

**Fig 6. Segments of the GoEmotions test set annotated automatically using the CoEQN framework.**

Fig 7 table:

| text | labels | admiration | amusement | anger | annoyance | approval | caring | confusion | curiosity | desire | disappoint | disapprov | disgust |
|---|---|---|---|---|---|---|---|---|---|---|---|---|---|
| I'm real | 25 | 0 | 0.22 | 0.81 | 0.31 | 1.38 | 0.47 | 0.39 | 0 | 1 | 2.12 | 0.58 | 1.12 |
| It's wond | 0 | 9.63 | 0.91 | 0 | 1.02 | 2.51 | 0.2 | 0.5 | 0 | 0 | 1.46 | 0.83 | 1.67 |
| Kings fan | 13 | 3.77 | 0 | 0 | 0 | 1.5 | 3.97 | 0 | 0 | 1.65 | 0 | 0 | 0 |
| I didn't | 15 | 0.45 | 0 | 0 | 0.2 | 0.83 | 0 | 0.62 | 0.7 | 0 | 0.11 | 0.2 | 0.08 |
| They got | 27 | 0 | 0 | 1.59 | 2.34 | 0.61 | 0.5 | 0.14 | 0 | 0.22 | 1.22 | 0.51 | 0.65 |
| Thank you | 15 | 1.06 | 0 | 0.63 | 0.27 | 0 | 0 | 0.96 | 0.26 | 0 | 0 | 0 | 0.15 |
| You're w | 15 | 1.9 | 0 | 0.05 | 0.35 | 2.45 | 0.75 | 0 | 0 | 0 | 0 | 0.66 | 0.84 |
| 100%! Cor | 15 | 7.19 | 0 | 0.87 | 0.48 | 0.27 | 0 | 0 | 0 | 0 | 0 | 0 | 0.7 |
| I'm sorr | 24 | 0 | 0 | 1.09 | 0.59 | 0.7 | 1.52 | 0.79 | 0 | 0.99 | 2.75 | 1.04 | 1.22 |
| Girlfrien | 25 | 2.55 | 0 | 3.46 | 4.71 | 2.98 | 0.02 | 0.16 | 0 | 0.41 | 3.34 | 2.53 | 3.49 |
| [NAME] ha | 3,10 | 0.43 | 0 | 0.64 | 1.29 | 3.58 | 0.79 | 0.31 | 0 | 0.12 | 0.66 | 1.8 | 0 |
| Lol! But | 1,18 | 1.13 | 7.23 | 0 | 0 | 0.01 | 0 | 0.3 | 0 | 0.86 | 0 | 0 | 0 |
| Translati | 8 | 0 | 0.29 | 1 | 0.72 | 0.17 | 1.2 | 0.52 | 1.93 | 5.49 | 0.77 | 0 | 0.03 |
| It's grea | 0,7 | 10 | 0.33 | 0 | 0 | 1.25 | 0.29 | 1.52 | 3.4 | 0 | 0.03 | 0.22 | 0.2 |
| I've also | 14 | 4.88 | 3.03 | 0 | 1.26 | 1.5 | 0 | 0.97 | 1.71 | 0.78 | 0.96 | 0.5 | 1.77 |
| I never w | 10 | 0 | 0.54 | 1.33 | 2.94 | 5.93 | 0.76 | 1.42 | 0 | 0.99 | 2.47 | 4.23 | 0.79 |
| The thoug | 14 | 0.37 | 0 | 3.28 | 4.52 | 3.17 | 0.4 | 0.42 | 0 | 0.36 | 2.53 | 2.84 | 3.85 |
| if the pa | 25,27 | 0 | 0 | 0.38 | 1.3 | 3.11 | 1.28 | 0.34 | 0.07 | 0.02 | 1.14 | 1.56 | 0.38 |
| Triggered | 1 | 0 | 10 | 0 | 0 | 0 | 0 | 0.37 | 0.81 | 0 | 0 | 0 | 0 |
| I'm autis | 15 | 2.98 | 0 | 0.22 | 0.57 | 1.13 | 0 | 0.17 | 0 | 0.03 | 0 | 0.42 | 0.08 |
| I know yo | 1 | 0 | 2.19 | 3.88 | 4.63 | 5.36 | 0.43 | 1.36 | 0.71 | 0.86 | 1.47 | 3.42 | 1.72 |
| May regre | 24 | 0 | 0 | 0.82 | 1.22 | 1.4 | 1 | 3.85 | 7.72 | 0.39 | 1.4 | 1.12 | 0.69 |
| After he | 27 | 0.18 | 0.38 | 0.37 | 0.85 | 0.59 | 0.24 | 0.71 | 1.83 | 0.32 | 0.83 | 0 | 0.13 |
| Well, the | 27 | 1.22 | 0 | 0.13 | 0.59 | 3.2 | 0.27 | 0.15 | 0.3 | 0.08 | 0.21 | 1.03 | 0.35 |
| Watch Veg | 6,27 | 0 | 0 | 0 | 1.26 | 2.92 | 0.47 | 1.54 | 1.57 | 0.41 | 1.03 | 1.44 | 0.5 |
| Again, ov | 27 | 0.72 | 0 | 0 | 0.89 | 6.8 | 0.6 | 0.69 | 0 | 0 | 0.84 | 3.96 | 0 |

**Fig 7. Segments of the GoEmotions test set annotated automatically using the EQN framework.**

**Table 4. Comparison of annotation results for the GoEmotionstestset using CoEQN and EQN.**

| class | Number of corpus | number of labels | CoEQNhits | CoEQNhit Rate | EQN hits | EQN hit rate |
|---|---|---|---|---|---|---|
| 1 label | 4590 | 4590 | 2416 | 0.5264 | 2534 | 0.5521 |
| 2 labels | 774 | 1548 | 565 | 0.3650 | 769 | 0.4968 |
| 3 labels | 61 | 183 | 83 | 0.4536 | 110 | 0.6011 |
| 4 labels | 2 | 8 | 5 | 0.6250 | 4 | 0.5000 |
| total | 5427 | 6329 | 3069 | 0.4849 | 3417 | 0.5399 |

**Table 5. Comparison of expanded experimental results.**

| class | number of Corpus | number of Labels | EQN hits | EQN hit rate |
|---|---|---|---|---|
| Top1 | 4590 | 4590 | 2416 | 0.5264 |
| Top2 | 4590 | 4590 | +761 | 0.6922 |
| Top3 | 4590 | 4590 | +401 | 0.7795 |

our framework possesses a commendable ability to uncover subtle emotional nuances in text. Further exploration in this area is left for the reader to contemplate and discover.

**Pearson coefficient heat map comparison.** To visually represent the results of the test set data annotated using the CoEQN and EQN frameworks, Pearson correlation coefficient heatmaps were generated based on the annotated intensity scores. These heatmaps, depicted in Figs 8 and 9, illustrate the degree of correlation among the various emotion labels. The data presented in the heatmaps effectively convey the relationships between the labels, providing insights into how different emotions are interconnected within the dataset.

A comparison of the two figures reveals that the Pearson correlation coefficients for the test set annotated using the EQN framework have largely increased, indicating a more pronounced correlation among the emotions. The interrelatedness of the 28 emotion categories reflects the nuances between them. Consequently, the adoption of the EQN framework provides more specific insights into the study of human emotional expression.

## Comparison of CoEQN and EQN annotation results with published results

In the development and application of machine learning models, evaluating the model's performance is a crucial step. Key indicators for assessing a model include Precision(prec), Recall, F1 score, macro-average, and standard deviation. When Google researchers released the GoEmotions dataset, they published relevant literature [13], which classified predictions on the GoEmotions test set based on the BERT model and provided statistical analysis and evaluation of the results.

To further evaluate the EQN framework, we similarly employed the BERT model and utilized the CoEQN and EQN frameworks described earlier to automatically annotate the energy scores for the 28 category labels of the GoEmotions test set. We computed the evaluation metrics for EQN based on the annotation results and conducted a comprehensive comparison with the results from the literature, thoroughly assessing the efficacy of the EQN framework.

Both experiments based on the EQN framework and the findings in literature [13] utilized the basic BERT model and the GoEmotions dataset. However, the literature does not provide detailed information regarding the basic parameter settings for BERT, which can be found in Section 2.2 of our study. The computed results for various evaluation metrics are compared with those presented in the literature and are summarized in Table 6.

The comparison table above reveals that, in contrast to the literature, the test results based on the EQN frameworks demonstrate varying degrees of improvement in precision and F1 scores across each label category. The F1-score serves

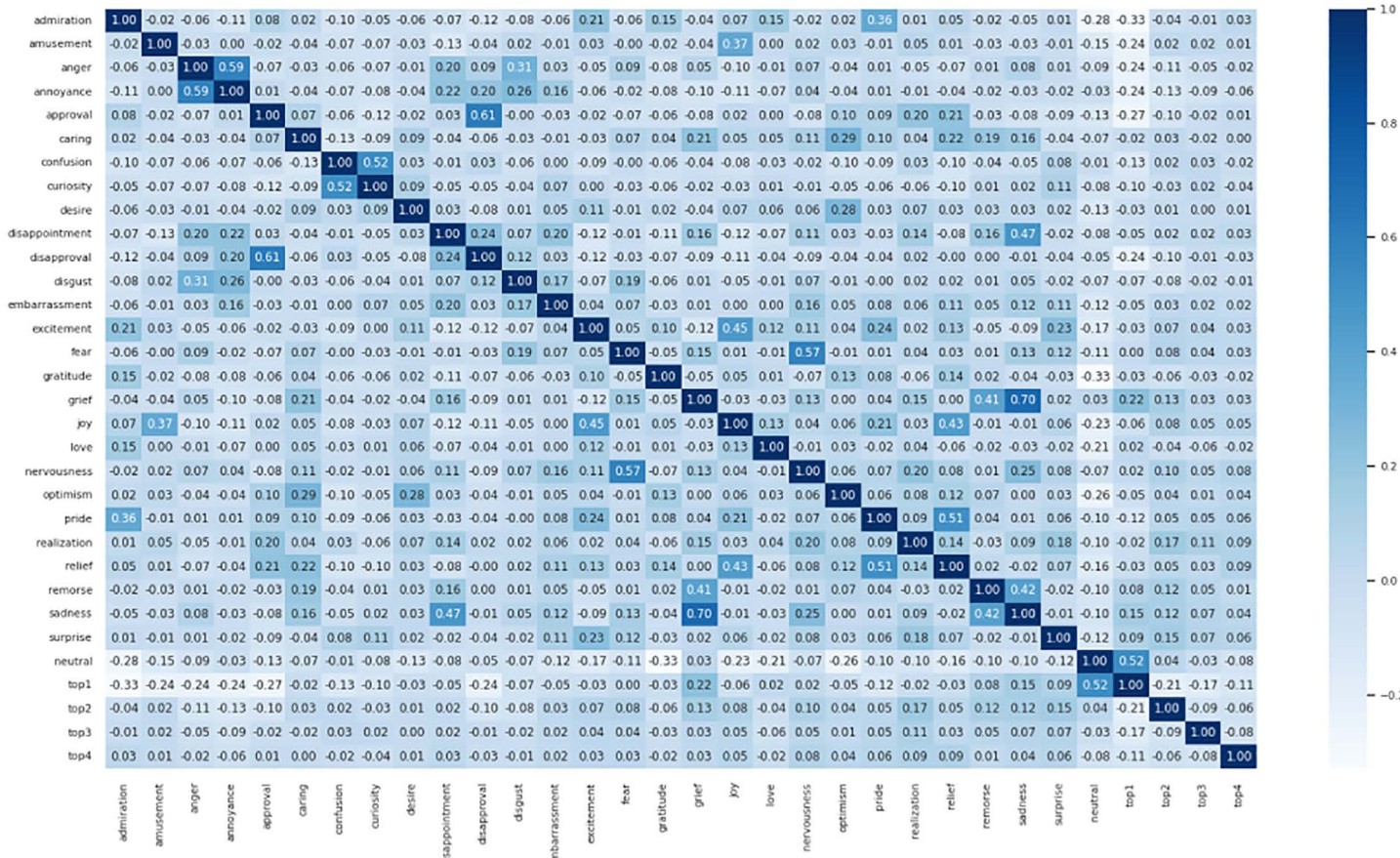

**Fig 8. Pearson correlation coefficient heatmap for the test set annotated using CoEQNframework.**

as a composite metric suitable for evaluating overall model performance, especially in the context of imbalanced class distributions.Our method improves the F1-scoreof 21 out of 28 categories.The macro-average of F1 score for the EQN experiment is 0.52, which is greater than the 0.50 achieved by the CoEQN experiment and also exceeds the literature score of 0.46. The Precision for the EQN framework stands at 0.56, which exceeds the 0.52 Precision achieved with the CoEQN framework, representing a 4% increase. Additionally, it surpasses the literature's Precision of 0.4 by 16%.. Moreover, the standard deviations for Precision, Recall, and F1-score across the two EQN framework experiments are lower than those reported in the literature, indicating that the results from the EQN frameworks are more stable. Our two models perform moderately in terms of Recall scores, with macro-average Recall of 0.49 and 0.51, which are lower than the literature.Overall, the result shows that both the CoEQN and EQN frameworks enhance the F1 and Precision, with the EQN framework yielding particularly stable and reliable results.

## Conclusion and outlook

This paper presents the extended quantitative EQN framework, designed to automate the annotation of micro-emotional datasets with emotional energy intensity scores. The framework is both straightforward and effective, addressing several critical challenges, including the high costs of manual annotation, subjective biases, significant

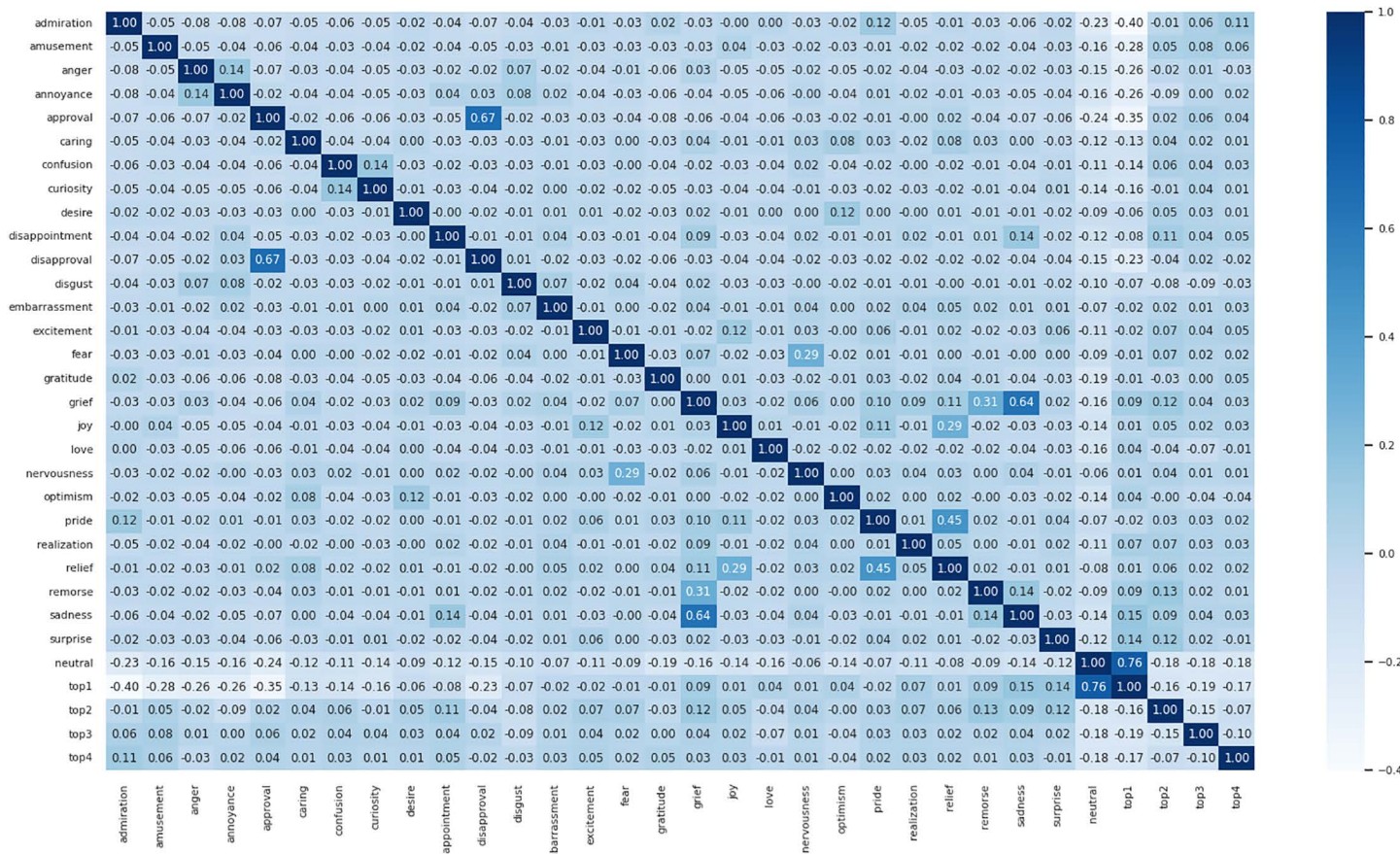

**Fig 9. Pearson correlation coefficient heatmap for the test set annotated using EQN framework.**

label imbalance, and limitations in annotating quantitative data. Furthermore, it enhances micro-emotional labeling within artificial datasets.

The EQN framework leverages the relationships among labels and employs a comprehensive numerical mapping method, which enables it to better capture the intricate complexities of the data and the interdependencies between labels. By effectively identifying unlabeled micro-emotional features within the original data, the framework significantly enhances the performance of automatic annotation and detection.

A series of extensive experiments utilizing multiple natural language processing (NLP) models were conducted to validate the framework's usability and generalizability. By employing the label regression training set, the EQN framework automatically annotated the GoEmotions training and testing sets with multi-label micro-emotions, complete with energy level scores, thereby enriching the GoEmotions dataset.

The continuous numerical representation of emotional energy levels provided by the automatic annotation process is particularly advantageous for quantitative emotional research. This innovation is expected to contribute positively to fields such as psychology and emotional computing, facilitating a more nuanced understanding of human emotions by machines.

Although this framework is based on micro-emotion labeling and detection, it is also applicable to text-related classification tasks.

**Table 6. Comparison of test results for CoEQN and EQN with literature [13].**

| Labels | Literature [13] | | | CoEQN | | | EQN | | |
|---|---|---|---|---|---|---|---|---|---|
| Emotion | Prec | Recall | F1 | Prec | Recall | F1 | Prec | Recall | F1 |
| admiration | 0.53 | 0.83 | 0.65 | 0.64 | 0.76 | 0.70 | 0.72 | 0.70 | 0.71 |
| amusement | 0.7 | 0.94 | 0.8 | 0.77 | 0.82 | 0.80 | 0.75 | 0.88 | 0.80 |
| anger | 0.36 | 0.66 | 0.47 | 0.48 | 0.48 | 0.48 | 0.54 | 0.49 | 0.52 |
| annoyance | 0.24 | 0.63 | 0.34 | 0.36 | 0.34 | 0.35 | 0.38 | 0.38 | 0.38 |
| approval | 0.26 | 0.57 | 0.36 | 0.32 | 0.44 | 0.37 | 0.25 | 0.53 | 0.34 |
| caring | 0.3 | 0.56 | 0.39 | 0.47 | 0.47 | 0.47 | 0.50 | 0.47 | 0.49 |
| confusion | 0.24 | 0.76 | 0.37 | 0.43 | 0.54 | 0.48 | 0.37 | 0.59 | 0.45 |
| curiosity | 0.4 | 0.84 | 0.54 | 0.52 | 0.49 | 0.51 | 0.54 | 0.51 | 0.52 |
| desire | 0.43 | 0.59 | 0.49 | 0.47 | 0.45 | 0.46 | 0.56 | 0.43 | 0.49 |
| disappointment | 0.19 | 0.52 | 0.28 | 0.38 | 0.30 | 0.34 | 0.37 | 0.31 | 0.34 |
| disapproval | 0.29 | 0.61 | 0.39 | 0.47 | 0.12 | 0.19 | 0.50 | 0.12 | 0.20 |
| disgust | 0.34 | 0.66 | 0.45 | 0.52 | 0.45 | 0.48 | 0.55 | 0.42 | 0.48 |
| embarrassment | 0.39 | 0.49 | 0.43 | 0.44 | 0.41 | 0.42 | 0.47 | 0.41 | 0.43 |
| excitement | 0.26 | 0.52 | 0.34 | 0.47 | 0.46 | 0.47 | 0.40 | 0.52 | 0.45 |
| fear | 0.46 | 0.85 | 0.6 | 0.61 | 0.67 | 0.64 | 0.61 | 0.77 | 0.68 |
| gratitude | 0.79 | 0.95 | 0.86 | 0.86 | 0.90 | 0.88 | 0.93 | 0.87 | 0.90 |
| grief | 0 | 0 | 0 | 0.25 | 0.17 | 0.20 | 1.00 | 0.33 | 0.50 |
| joy | 0.39 | 0.73 | 0.51 | 0.66 | 0.57 | 0.61 | 0.58 | 0.69 | 0.63 |
| love | 0.68 | 0.92 | 0.78 | 0.80 | 0.79 | 0.80 | 0.79 | 0.83 | 0.81 |
| nervousness | 0.28 | 0.48 | 0.35 | 0.37 | 0.30 | 0.33 | 0.38 | 0.39 | 0.38 |
| neutral | 0.56 | 0.84 | 0.68 | 0.50 | 0.55 | 0.52 | 0.62 | 0.48 | 0.54 |
| optimism | 0.41 | 0.69 | 0.51 | 0.55 | 0.38 | 0.44 | 0.71 | 0.31 | 0.43 |
| pride | 0.67 | 0.25 | 0.36 | 0.34 | 0.28 | 0.30 | 0.27 | 0.27 | 0.27 |
| realization | 0.16 | 0.29 | 0.21 | 0.38 | 0.27 | 0.32 | 0.60 | 0.27 | 0.38 |
| relief | 0.5 | 0.09 | 0.15 | 0.61 | 0.75 | 0.67 | 0.58 | 0.63 | 0.60 |
| remorse | 0.53 | 0.88 | 0.66 | 0.54 | 0.53 | 0.54 | 0.57 | 0.58 | 0.58 |
| sadness | 0.38 | 0.71 | 0.49 | 0.62 | 0.48 | 0.54 | 0.55 | 0.60 | 0.57 |
| surprise | 0.4 | 0.66 | 0.5 | 0.63 | 0.65 | 0.64 | 0.66 | 0.57 | 0.61 |
| macro-average | 0.4 | 0.63 | 0.46 | 0.52 | 0.49 | 0.50 | 0.56 | 0.51 | 0.52 |
| std | 0.18 | 0.24 | 0.19 | 0.15 | 0.19 | 0.17 | 0.17 | 0.19 | 0.16 |

## Supporting information

**S1 File. Pseudocode for the Python Implementation of the EQN Framework (Using the BERT Model as an Example).**
(PDF)

## Acknowledgments

This research is partially supported by the 242 National InformationSecurity Projects, PR China under Grant 2020A065.

## Author contributions

**Funding acquisition:** Senlin Luo.

**Investigation:** Haofan Chen.

**Software:** Jingyi Zhou.

**Supervision:** Senlin Luo.

**Validation:** Haofan Chen.

**Visualization:** Haofan Chen.

**Writing – original draft:** Jingyi Zhou.

**Writing – review & editing:** Jingyi Zhou, Senlin Luo.

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
