## [Decision Letter · Decision Letter 0]

22 Feb 2025

Dear Dr. Zhou,

Thank you for submitting your manuscript to PLOS ONE. After careful consideration, we feel that it has merit but does not fully meet PLOS ONE’s publication criteria as it currently stands. Therefore, we invite you to submit a revised version of the manuscript that addresses the points raised during the review process.

We look forward to receiving your revised manuscript.

Kind regards,

Alemayehu Getahun Kumela, Ph.D.

Academic Editor

PLOS ONE

This research is partially supported by the 242 National Information Security Projects,PR China under Grant 2020A065.

This research is partially supported by the 242 National Information Security Projects, PR China under Grant 2020A065.

This research is partially supported by the 242 National Information Security Projects,PR China under Grant 2020A065.

5. We are unable to open your Supporting Information file plos_latex_template.tex. Please kindly revise as necessary and re-upload.

Reviewers' comments:

Reviewer's Responses to Questions

**Comments to the Author**

1. Is the manuscript technically sound, and do the data support the conclusions?

Reviewer #1: Yes

Reviewer #2: Yes

2. Has the statistical analysis been performed appropriately and rigorously?

Reviewer #1: Yes

Reviewer #2: Yes

3. Have the authors made all data underlying the findings in their manuscript fully available?

Reviewer #1: Yes

Reviewer #2: Yes

4. Is the manuscript presented in an intelligible fashion and written in standard English?

Reviewer #1: Yes

Reviewer #2: No

Reviewer #1: The article you've provided, titled Expansion Quantization Network: A Micro-emotion Detection and Annotation Framework, is focused on the development of an innovative method for detecting and annotating micro-emotions in text. The framework proposed, called EQN (Emotion Quantization Network), aims to address several challenges in the field of emotion detection, including label imbalance, high annotation costs, and subjectivity in labeling.

Here are some areas where the paper might have potential weaknesses or areas for improvement:

1. While the paper compares the EQN framework with various NLP models and methods, it does not provide sufficient context on the baseline methods or the most up-to-date state-of-the-art models in the field. The authors could improve their comparative analysis by more explicitly benchmarking the EQN framework against the most recent advancements in emotion detection and annotation, such as recent work involving large language models or newer techniques in emotion classification.

2. The EQN framework, although novel, seems quite complex with multiple steps such as full-label initialization, regression, and training set annotation. The method may not be easily replicable for other researchers or practitioners. Simplifying the explanation or providing a more detailed, step-by-step guide with pseudocode could help in increasing the accessibility and usability of the framework. More emphasis could be placed on how the framework can be adapted or scaled to other datasets or applications.

3. The datasets used for validation (e.g., GoEmotions, 7health, 6emotions) seem to be limited in terms of diversity in emotion categories, especially with certain categories being underrepresented. Including additional, more diverse datasets or highlighting more challenging datasets would better demonstrate the robustness and generalizability of the EQN framework. The paper could also discuss how the framework can handle highly imbalanced datasets or edge cases more effectively.

4. While BERT is a powerful model for text understanding, its heavy reliance could raise concerns about the general applicability of the framework to other models or domains outside of those primarily using BERT. The authors should discuss how the EQN framework performs with different models, not only BERT, and whether it could be generalized across other NLP tasks (such as sentiment analysis or topic modeling). More variety in the tested models would also highlight the framework’s flexibility.

5. The paper claims to address micro-emotion detection but lacks a thorough explanation of how micro-emotions are defined or what specific examples they cover. This vagueness may confuse readers or limit the paper’s broader application in emotion analysis. A more detailed definition and examples of micro-emotions in text would clarify their role within the EQN framework. The paper could benefit from a deeper exploration of the specific emotional categories used, especially how subtle emotional states are quantified and detected.

6. The paper does not address the computational cost or efficiency of implementing the EQN framework. With complex frameworks often requiring significant resources, this omission could hinder real-world deployment. The authors should include a discussion on the computational complexity of their method, including training times, required hardware, and potential bottlenecks. Providing a performance breakdown would be valuable for practitioners considering implementing the framework in real-time applications.

7. The results show significant improvements in precision, recall, and F1-score across categories, but the paper does not thoroughly discuss potential overfitting, especially on datasets with a smaller number of categories. The authors should perform cross-validation or provide additional robustness checks to ensure that the performance improvements are not simply overfitting to specific datasets. Discussing how the EQN framework generalizes across various domains or datasets would strengthen the claims of its broad applicability.

8. The evaluation is mostly quantitative, with metrics like precision, recall, and F1-score dominating the discussion. However, these metrics do not always capture the nuance of micro-emotion detection. Including a qualitative analysis of the framework’s ability to detect nuanced emotions (e.g., examples of text annotated by the framework versus human annotators) would provide deeper insights into its practical capabilities and limitations.

9. Suggestion references:

- S. Saifullah, R. Dreżewski, F. A. Dwiyanto, A. S. Aribowo, Y. Fauziah, and N. H. Cahyana, “Automated Text Annotation Using a Semi-Supervised Approach with Meta Vectorizer and Machine Learning Algorithms for Hate Speech Detection,” Appl. Sci., vol. 14, no. 3, p. 1078, Jan. 2024, doi: 10.3390/app14031078.

- Saifullah, S., Dreżewski, R., Dwiyanto, F.A., Aribowo, A.S., Fauziah, Y. (2023). Sentiment Analysis Using Machine Learning Approach Based on Feature Extraction for Anxiety Detection. In: Mikyška, J., de Mulatier, C., Paszynski, M., Krzhizhanovskaya, V.V., Dongarra, J.J., Sloot, P.M. (eds) Computational Science – ICCS 2023. ICCS 2023. Lecture Notes in Computer Science, vol 14074. Springer, Cham. https://doi.org/10.1007/978-3-031-36021-3_38

Reviewer #2: Dear Editor,

In this manuscript, the authors explore the Expansion Quantization Network, a framework for micro-emotion detection and annotation. This straightforward and effective framework addresses critical challenges such as cost, subjective biases, label imbalance, and limitations in quantitative annotation, while also enhancing micro-emotional labeling within artificial datasets.

Therefore, in my opinion, the results presented in the paper are both reasonable and intriguing, making them worthy of publication in the Journal of PLOS ONE following minor revisions. However, a few aspects should be addressed to enhance the quality of the paper, including:

1. Improving the clarity and depth of the statement of the problem.

2. The quality of the figures and tables could be enhanced by utilizing tools for vector images. I recommend using open-source software such as Inkscape or GIMP for this purpose.

3. Enhancing the conclusion according to your results.

4. The governing equations should be appropriately cited.

5. Your document contains several typos and mistakes. I recommend that the authors review the English text using tools such as the open version of Grammarly.

**Do you want your identity to be public for this peer review?** For information about this choice, including consent withdrawal, please see our Privacy Policy

Reviewer #1: **Yes: ** Shoffan Saifullah

Reviewer #2: No

---

## [Author Response · Author response to Decision Letter 1]

25 Mar 2025

Thank you for taking the time to review my manuscript and for providing valuable feedback. In response to your suggestions, I have thoroughly revised the paper and made improvements in various areas. Regarding the issue with the supporting information file, the revised submission includes a Word version consistent with the original LaTeX version, and the corresponding figures have been separately uploaded according to the publication requirements.

Below is a detailed response to each reviewer’s comments, outlining the modifications made and the corresponding explanations. I hope these revisions will further enhance the quality of the paper and meet the journal's requirements.

Reviewer #1:

1.While the paper compares the EQN framework with various NLP models and methods, it does not provide sufficient context on the baseline methods or the most up-to-date state-of-the-art models in the field. The authors could improve their comparative analysis by more explicitly benchmarking the EQN framework against the most recent advancements in emotion detection and annotation, such as recent work involving large language models or newer techniques in emotion classification.

Response:

Dear Expert,Thank you for your valuable suggestions. In the "Related Work" section of the original manuscript, we introduced the study by WANG Yaoqi et al. (2024) [27] and the research on the EmoLLMs dataset based on ChantGpt [31], which represent the latest automated annotation methods for large language models or emotion classification.

In response to your recommendations, we have further supplemented the "Related Work" section in the revised manuscript with the latest research findings on emotion classification. Please refer to lines 122–162 of the revised version for details.

2.The EQN framework, although novel, seems quite complex with multiple steps such as full-label initialization, regression, and training set annotation. The method may not be easily replicable for other researchers or practitioners. Simplifying the explanation or providing a more detailed, step-by-step guide with pseudocode could help in increasing the accessibility and usability of the framework. More emphasis could be placed on how the framework can be adapted or scaled to other datasets or applications.

Response:

Thank you for recognizing our work and for pointing out its shortcomings. Following your suggestion, we have added supplementary information at the end of the paper: the pseudocode for the Python implementation of the EQN framework. This file (S1_EQN-python-code.pdf) has been uploaded along with the revised manuscript. This addition aims to help readers quickly understand and replicate our framework.

Additionally, we have included the statement "Our EQN framework is applicable to manually annotated single-label or multi-label emotion datasets." before the "Contributions of This Paper" section in the revised manuscript.

3. The datasets used for validation (e.g., GoEmotions, 7health, 6emotions) seem to be limited in terms of diversity in emotion categories, especially with certain categories being underrepresented. Including additional, more diverse datasets or highlighting more challenging datasets would better demonstrate the robustness and generalizability of the EQN framework. The paper could also discuss how the framework can handle highly imbalanced datasets or edge cases more effectively.

Response:

Dear Expert,

Thank you very much for your valuable suggestions. Regarding the diversity of emotion categories, we have made extensive efforts to identify datasets from multiple perspectives. We selected datasets covering seven health-related emotions, six fundamental human emotions, three socio-economic emotions, three categories of emotions in online user posts, and a human emotion dataset with 28 fine-grained categories. Based on the results from these datasets, the robustness and generalizability of the EQN framework have been largely validated.

Using our framework, we have released a Chinese Weibo dataset annotated with sentiment scores at https://github.com/yeaso/Chinese-Affective-Computing-Dataset.

Regarding the effective handling of highly imbalanced datasets, this remains a critical topic in machine learning research. As a general framework, our approach incorporates traditional oversampling and undersampling techniques in the preprocessing stage to balance the dataset. The original manuscript primarily focused on differentiating EQN from traditional methods and validating its effectiveness.

Following your suggestion, we have added supplementary information at the end of the paper: the pseudocode for the Python implementation of the EQN framework, which includes a detailed introduction to how imbalanced datasets are handled in the preprocessing stage.

4. While BERT is a powerful model for text understanding, its heavy reliance could raise concerns about the general applicability of the framework to other models or domains outside of those primarily using BERT. The authors should discuss how the EQN framework performs with different models, not only BERT, and whether it could be generalized across other NLP tasks (such as sentiment analysis or topic modeling). More variety in the tested models would also highlight the framework’s flexibility.

Response:

Dear Expert,

We appreciate your concern and would like to provide an explanation. In our framework, the model component can be any machine learning model; however, different models exhibit varying performance in text classification. In the experimental section, we conducted comparative tests using ANN, CNN, LSTM, TextCNN, and BERT on five datasets. The results demonstrate that the EQN framework is applicable across all models, confirming its generalizability. Among them, BERT performed exceptionally well, aligning with the expected strong advantages of large language models in text classification. Consequently, we selected BERT for the automatic annotation of the GoEmotions dataset.

When the GoEmotions dataset was manually annotated and released, the Google team also used BERT for statistical analysis of the dataset. By comparing our EQN framework’s automatic annotation results with theirs, we further validated the robustness of our framework in automatic annotation.

Although the primary goal of the EQN framework is micro-emotion annotation, its core mechanism is based on text classification, making it equally applicable to automatic annotation of macro-emotions and related text classification tasks, such as sentiment analysis and topic classification.

5. The paper claims to address micro-emotion detection but lacks a thorough explanation of how micro-emotions are defined or what specific examples they cover. This vagueness may confuse readers or limit the paper’s broader application in emotion analysis. A more detailed definition and examples of micro-emotions in text would clarify their role within the EQN framework. The paper could benefit from a deeper exploration of the specific emotional categories used, especially how subtle emotional states are quantified and detected.

Response:

Dear Expert,

Since publicly available emotion datasets are primarily manually annotated, it is challenging to label micro-emotions, which is why research on micro-emotions in the literature remains scarce. In the background section of our paper, we provide a detailed description of micro-expressions and micro-emotions. Specifically, we describe micro-emotions as follows:

"While macro-expressions provide relatively straightforward and direct representations of emotions, micro-expressions more accurately reflect subtle, unconscious, or fleeting emotional states."

Our EQN framework is specifically designed to quantify and detect subtle emotional states. All experimental examples in this paper focus on the quantitative analysis and evaluation of both emotions and micro-emotions. We hope this explanation helps address your concerns.

6. The paper does not address the computational cost or efficiency of implementing the EQN framework. With complex frameworks often requiring significant resources, this omission could hinder real-world deployment. The authors should include a discussion on the computational complexity of their method, including training times, required hardware, and potential bottlenecks. Providing a performance breakdown would be valuable for practitioners considering implementing the framework in real-time applications.

Response:

Dear Expert,

The EQN framework is a general framework based on standard models, and the cost or efficiency primarily depends on the specific model used. In the experimental section, we evaluate five fundamental models: ANN, CNN, LSTM, TextCNN, and BERT. The cost and efficiency metrics can be referenced from the respective figures published by the original developers of these models.

When applying this framework, users can select an appropriate model based on their specific application requirements.

7. The results show significant improvements in precision, recall, and F1-score across categories, but the paper does not thoroughly discuss potential overfitting, especially on datasets with a smaller number of categories. The authors should perform cross-validation or provide additional robustness checks to ensure that the performance improvements are not simply overfitting to specific datasets. Discussing how the EQN framework generalizes across various domains or datasets would strengthen the claims of its broad applicability.

Response:

Dear Expert,

We would like to clarify this issue. The EQN framework maps single-label and multi-label samples to a full-label representation with energy-level scores, effectively utilizing label dependencies. In the output stage, the framework generates predictions for all categories and ultimately produces either single-label multi-class or multi-label multi-class results based on task requirements. This unified framework addresses label sparsity and imbalance issues in multi-label tasks, thereby improving model prediction performance to some extent.

In our framework, both input and output labels are represented as continuous, quantified values, making it analogous to stock price prediction or real estate price forecasting systems—essentially a regression problem. Therefore, EQN serves as a regression-based framework rather than a traditional text classification approach.

Regarding the comment that "the authors should conduct cross-validation or provide additional robustness checks," we have applied cross-validation in all model training experiments and reported the average evaluation results. Since cross-validation is a fundamental methodology in model training, we did not provide a detailed explanation in the paper.

8. The evaluation is mostly quantitative, with metrics like precision, recall, and F1-score dominating the discussion. However, these metrics do not always capture the nuance of micro-emotion detection. Including a qualitative analysis of the framework’s ability to detect nuanced emotions (e.g., examples of text annotated by the framework versus human annotators) would provide deeper insights into its practical capabilities and limitations.

Response:

Dear Expert,

In the paper, the EQN framework was trained on five models and five datasets, with evaluation comparisons using metrics such as accuracy, recall, and F1-score, demonstrating the framework's generalizability and adaptability. To further validate the practical applicability of EQN, we selected the most challenging manually annotated 28-class emotion dataset, GoEmotions, for testing.

The experimental results in Table 6. Comparison of Test Results for CoEQN and EQN with Literature show that the EQN framework's automatic annotation for the 28 emotion categories is close to human classification levels, with some metrics even surpassing human classification performance. The label regression method proposed in this paper further proves that the full-label approach can effectively capture micro-features not annotated in the original data, significantly improving model performance.

With this method, using small-scale, highly accurate manual annotations and high-quality models, it is possible to achieve high-level machine annotation for large-scale emotion datasets.

9. Suggestion references:

- S. Saifullah, R. Dreżewski, F. A. Dwiyanto, A. S. Aribowo, Y. Fauziah, and N. H. Cahyana, “Automated Text Annotation Using a Semi-Supervised Approach with Meta Vectorizer and Machine Learning Algorithms for Hate Speech Detection,” Appl. Sci., vol. 14, no. 3, p. 1078, Jan. 2024, doi: 10.3390/app14031078.

- Saifullah, S., Dreżewski, R., Dwiyanto, F.A., Aribowo, A.S., Fauziah, Y. (2023). Sentiment Analysis Using Machine Learning Approach Based on Feature Extraction for Anxiety Detection. In: Mikyška, J., de Mulatier, C., Paszynski, M., Krzhizhanovskaya, V.V., Dongarra, J.J., Sloot, P.M. (eds) Computational Science – ICCS 2023. ICCS 2023. Lecture Notes in Computer Science, vol 14074. Springer, Cham. https://doi.org/10.1007/978-3-031-36021-3_38

Response:

Thank you for providing the latest research related to this paper. After careful review, we believe it is very helpful in enriching the content of the paper. We have made additions to the "Related Work" section in the revised manuscript. Please refer to references 25 and 28 in the References section.

Reviewer #2: Dear Editor,

In this manuscript, the authors explore the Expansion Quantization Network, a framework for micro-emotion detection and annotation. This straightforward and effective framework addresses critical challenges such as cost, subjective biases, label imbalance, and limitations in quantitative annotation, while also enhancing micro-emotional labeling within artificial datasets.

Therefore, in my opinion, the results presented in the paper are both reasonable and intriguing, making them worthy of publication in the Journal of PLOS ONE following minor revisions. However, a few aspects should be addressed to enhance the quality of the paper, including:

Response:

Dear Expert,

We sincerely appreciate the reviewer’s high recognition of our work. We have carefully and diligently made revisions based on the reviewer’s suggestions to meet the journal’s publication standards.

1. Improving the clarity and depth of the statement of the problem.

Response:

Dear Expert,

Thank you for this helpful suggestion. To assist readers in quickly understanding the structure and workflow of the EQN framework, we have added supplementary information at the end of the paper: EQN Framework Python Implementation Pseudocode (using the BERT model as an example).

2. The quality of the figures and tables could be enhanced by utilizing tools for vector images. I recommend using open-source software such as Inkscape or GIMP for this purpose.

Response:

Dear Expert,

You are correct that the quality of the figures in the original manuscript was somewhat blurry. Thank you for recommending the image processing tool. In the revised manuscript, we have reprocessed all the images using Inkscape, which has significantly improved the quality of the figures and the overall paper.

3. Enhancing the conclusion according to your results.

Response:

Thank you for your suggestion. We have added the following to the conclusion section: “Although this framework is based on micro-emotion labeling and detection, it is also applicable to text-related classification tasks.”

4. The governing equations should be appropriately cited.

Response:

Thank you for the suggestions. We would like to provide some clarification. Formulas 1-3 are algorithms used for data preprocessing in the EQN framework, and they have been referenced in the newly added pseudocode. Formulas 4-8 are algorithms for processing data within the EQN framework model. Formulas 9-15 are the calculation formulas for evaluating multiple performance metrics of the EQN framework, and the EQN metrics presented in the "Experimental" section of the paper are derived using the algorithms from formulas 9-15.

5. Your document contains several typos and mistakes. I recommend that the authors review the English text using tools such as the open version of Grammarly.

Response:

Thank you very much for rec

---

## [Decision Letter · Decision Letter 1]

22 Jun 2025

Dear Dr. Jingyi Zhou,

Thank you for submitting your manuscript to PLOS ONE. After careful consideration, we feel that it has merit but does not fully meet PLOS ONE’s publication criteria as it currently stands. Therefore, we invite you to submit a revised version of the manuscript that addresses the points raised during the review process.

We look forward to receiving your revised manuscript.

Kind regards,

Alemayehu Getahun Kumela, Ph.D.

Academic Editor

PLOS ONE

Journal Requirements:

Reviewers' comments:

Reviewer's Responses to Questions

**Comments to the Author**

Reviewer #2: All comments have been addressed

Reviewer #3: (No Response)

2. Is the manuscript technically sound, and do the data support the conclusions?

Reviewer #2: Yes

Reviewer #3: (No Response)

3. Has the statistical analysis been performed appropriately and rigorously?

Reviewer #2: Yes

Reviewer #3: Yes

4. Have the authors made all data underlying the findings in their manuscript fully available?

Reviewer #2: Yes

Reviewer #3: Yes

5. Is the manuscript presented in an intelligible fashion and written in standard English?

Reviewer #2: Yes

Reviewer #3: Yes

Reviewer #2: (No Response)

Reviewer #3: Some small things should still be addressed/discussed:

First, the EQN pipeline-full-label initialization, regression, and training-set annotation-has so many pieces that other researchers may struggle to set it up exactly as written.

Second, the validation sets- GoEmotions, Thealth, Gemotions-cover few emotion classes, leaving some feelings seriously underrepresented. That gap weakens claims about robustness

Third, because BERT carries most of the load (model dependency), I wonder whether the approach transfers to models like ANN, CNN, LSTM, or TextCNN that the team did test. BERTs dominance still hints at a narrow sweet spot.

Last, despite the title, the paper says little about what micro-emotions are or gives concrete examples. This ambiguity may puzzle readers and, in turn, shrink the papers impact.

**Do you want your identity to be public for this peer review?** For information about this choice, including consent withdrawal, please see our Privacy Policy

Reviewer #2: No

Reviewer #3: No

---

## [Author Response · Author response to Decision Letter 2]

27 Jun 2025

Thank you for taking the time to review my manuscript and for providing valuable feedback.

Below is a detailed response to reviewer’s comments, outlining the modifications made and the corresponding explanations. I hope these revisions will further enhance the quality of the paper and meet the journal's requirements.

Reviewer #3: Some small things should still be addressed/discussed:

First, the EQN pipeline-full-label initialization, regression, and training-set annotation-has so many pieces that other researchers may struggle to set it up exactly as written.

Response: We appreciate the reviewer’s insightful feedback. Indeed, the original description of the EQN process was somewhat verbose and may have caused confusion. Now, we have revised the corresponding section of the manuscript (see Lines 266–289 in the revised version). The updated description is now clearer, more structured, and more accessible for readers to follow.

The specific revisions are as follows:

The EQN framework consists of a two-stage training pipeline for enhancing emotion classification through full-label regression. Its operation is straightforward and can be summarized as follows:

1. Data Preparation: Each training sample with a single emotion label is transformed into a multi-dimensional one-hot vector representing the full emotion label space (e.g., for 28 emotions in GoEmotions).

2. Stage 1 – Full Label Initialization & Model A Training:

A base classification model (Model A) is trained on the initialized dataset using standard loss functions (e.g., MSE loss for regression).

Model A learns to map input texts to emotion label vectors.

3. Soft Label Generation:

Model A is used to predict probability distributions over the entire label space for each training sample.

These predicted probabilities serve as soft labels for all emotions not originally annotated (while preserving the original label as 1.0).

4. Stage 2 – Model B Training with Refined Labels:

A second model (Model B) is trained on the soft-labeled dataset to learn a more robust and generalized representation of emotional features.

5. Inference:

Model B can be used to classify emotions of new texts or assign multi-dimensional emotional scores for downstream tasks.

Note: Steps 1–3 are referred to as the core EQN module (CoEQN), which is also used as a standalone component in Section 4 for ablation experiments.

A full Python-style pseudocode is provided in the Supplementary Material to ensure reproducibility.

We believe this clearer description, along with the accompanying pseudocode, ensures that researchers can implement the EQN framework without difficulty.

Second, the validation sets- GoEmotions, Thealth, Gemotions-cover few emotion classes, leaving some feelings seriously underrepresented. That gap weakens claims about robustness.

Response: Thank you for your valuable comment. We fully understand your concern regarding the limited coverage of emotion categories in the validation datasets.Our study has deliberately incorporated a diverse range of datasets that span various levels of emotional granularity (3, 6, 7, and 28 classes), ensuring both breadth and representativeness in our robustness evaluation of the EQN framework. These datasets include a wide range of emotional domains, from basic emotions to fine-grained states, and cover multiple real-world application areas. Specifically, we used the following datasets:

• 7health dataset: Related to mental health, containing 51,074 entries labeled with seven categories—anxiety, bipolar disorder, depression, normal, personality disorder (PD), stress, and suicidal ideation.

• 6emotions dataset: An English corpus annotated with six basic emotions—sadness, joy, love, anger, fear, and surprise.

• 3TFN dataset: Financial domain tweets annotated for sentiment polarity.

• 3TSA dataset: Approximately 163,000 tweets labeled with three sentiment categories—positive, neutral, and negative.

• GoEmotions dataset: Released by Google, featuring 28 fine-grained emotion labels. It is one of the most detailed and comprehensive publicly available emotion datasets.

By combining these datasets, our study covers a wide range of emotional contexts—including human affective states, mental health, financial sentiment, and social media emotion analysis—with both coarse- and fine-grained labels. Therefore, we believe the robustness evaluation of EQN is well-grounded and broadly representative.

Third, because BERT carries most of the load (model dependency), I wonder whether the approach transfers to models like ANN, CNN, LSTM, or TextCNN that the team did test. BERTs dominance still hints at a narrow sweet spot.

Response: We appreciate the reviewer’s thoughtful concern regarding potential model dependency in our approach. Our proposed framework EQN is designed to be model-agnostic, and we thoroughly tested it across five diverse classification architectures: ANN, CNN, TextCNN, LSTM, and BERT.

To directly address the concern of model transferability, we present in Table 3 a comprehensive comparison of three classification strategies—conventional single-label (S), multi-label (M), and our EQN full-label mapping approach (EQN)—evaluated on five datasets. The results demonstrate that EQN consistently outperforms both S and M baselines across all tested models, including those with no pretrained components (ANN, CNN, TextCNN, LSTM, BERT).

Table 3.Testing results of single-label, multi-label, and full-label mapping classification models.

model/dataset 7health 6emotions 3TFN 3TSA 28GoEmotions

ANN S 0.3202 0.3358 0.6557 0.5199

M 　 　 　 　 0.1611

EQN 0.374 0.3551 0.6482 0.5254 0.1877

CNN S 0.6658 0.6778 0.7353 0.8166 　

M 　 　 　 　 0.3892

EQN 0.6836 0.7113 0.7592 0.8161 0.4076

TextCNN S 0.6624 0.7603 0.7575 0.7357 　

M 　 　 　 　 0.3889

EQN 0.6672 0.7869 0.7717 0.7448 0.4205

LSTMS S 0.7068 0.9318 0.8099 0.93 　

M 　 　 　 　 0.4545

EQN 0.7295 0.9352 0.8073 0.9283 0.4658

BERT S 0.7855 0.9328 0.8378 0.9405 　

M 　 　 　 　 0.4665

EQN 0.8034 0.935 0.8458 0.9413 0.4849

Table notation: single-label is denoted as S, multi-label as M, and EQN full label mapping as EQN.

An illustrative example is provided based on the experimental results in Table 3.:

• On the GoEmotions dataset, EQN improves performance over multi-label classification by:

+0.0266 on CNN (from 0.3892 to 0.4076),

+0.0316 on TextCNN (from 0.3889 to 0.4205),

+0.0113 on LSTM (from 0.4545 to 0.4658),

+0.0184 on BERT (from 0.4665 to 0.4849).

• On 6emotions, EQN outperforms single-label classification by:

+0.0193 on ANN,

+0.0335 on CNN,

+0.0266 on TextCNN,

+0.0034 on LSTM,

+0.0022 on BERT.

The analysis of the above data indicates that, from the perspective of relative performance improvement, BERT does not have an advantage over other models after applying the EQN framework. The reason for choosing BERT is that it generally achieves better performance than other models in natural language processing tasks. This supports our claim that EQN is broadly applicable and not confined to a narrow sweet spot.

Last, despite the title, the paper says little about what micro-emotions are or gives concrete examples. This ambiguity may puzzle readers and, in turn, shrink the papers impact.

Response:

Thank you for your insightful comment. In the revised manuscript, we have added a detailed explanation and concrete examples of micro-emotions in the Introduction section (lines 107–117) to improve clarity and reduce potential reader confusion.

The supplementary content added in the revised manuscript is as follows:

“In the context of natural language, micro-emotions refer to fleeting, low-intensity, and often subconscious emotional states expressed subtly through text. These emotions are typically harder to detect than macro-emotions such as joy or anger, as they are conveyed through nuanced wording, implicit sentiment, or slight linguistic variations. For instance, a sentence like "That's exactly the kind of brilliant nonsense I expected" may carry sarcasm mixed with disappointment and helplessness—emotional shades that would be missed by coarse labeling systems. Micro-emotions provide a deeper view into the speaker’s internal psychological state, making them highly valuable in fields such as sentiment analysis, mental health screening, and user experience optimization. Our study considers micro-emotions as integral components of fine-grained emotion modeling and aims to capture them systematically through quantification and annotation.”

---

## [Decision Letter · Decision Letter 2]

22 Sep 2025

Expansion quantization network: A micro-emotion detection and annotation framework

PONE-D-24-58714R2

Dear Dr. Jingyi Zhou,

We’re pleased to inform you that your manuscript has been judged scientifically suitable for publication and will be formally accepted for publication once it meets all outstanding technical requirements.

Kind regards,

Alemayehu Getahun Kumela, Ph.D.

Academic Editor

PLOS ONE

Additional Editor Comments (optional):

Reviewer #3:

Reviewers' comments:

Reviewer's Responses to Questions

**Comments to the Author**

Reviewer #3: All comments have been addressed

2. Is the manuscript technically sound, and do the data support the conclusions?

Reviewer #3: Yes

3. Has the statistical analysis been performed appropriately and rigorously?

Reviewer #3: Yes

4. Have the authors made all data underlying the findings in their manuscript fully available?

Reviewer #3: Yes

5. Is the manuscript presented in an intelligible fashion and written in standard English?

Reviewer #3: Yes

Reviewer #3: The authors present with with scientific quality the anwers to the questions presented. All the main issues were adressed.

**Do you want your identity to be public for this peer review?** For information about this choice, including consent withdrawal, please see our Privacy Policy

Reviewer #3: **Yes: ** João M.F. Rodrigues

---

## [Editor Report · Acceptance letter]

PONE-D-24-58714R2

PLOS ONE

Dear Dr. Zhou,

I'm pleased to inform you that your manuscript has been deemed suitable for publication in PLOS ONE. Congratulations! Your manuscript is now being handed over to our production team.

Kind regards,

on behalf of

Dr. Alemayehu Getahun Kumela

Academic Editor

PLOS ONE